# Titanium-based potassium-ion battery positive electrode with extraordinarily high redox potential

Stanislav S. Fedotov [1✉], Nikita D. Luchinin [2], Dmitry A. Aksyonov [1], Anatoly V. Morozov[1], Sergey V. Ryazantsev [1,2], Mattia Gaboardi [3], Jasper R. Plaisier [3], Keith J. Stevenson [1], Artem M. Abakumov[1] & Evgeny V. Antipov [1,2]

The rapid progress in mass-market applications of metal-ion batteries intensifies the development of economically feasible electrode materials based on earth-abundant elements. Here, we report on a record-breaking titanium-based positive electrode material, $KTiPO_4F$, exhibiting a superior electrode potential of 3.6 V in a potassium-ion cell, which is extraordinarily high for titanium redox transitions. We hypothesize that such an unexpectedly major boost of the electrode potential benefits from the synergy of the cumulative inductive effect of two anions and charge/vacancy ordering. Carbon-coated electrode materials display no capacity fading when cycled at 5C rate for 100 cycles, which coupled with extremely low energy barriers for potassium-ion migration of 0.2 eV anticipates high-power applications. Our contribution shows that the titanium redox activity traditionally considered as "reducing" can be upshifted to near-4V electrode potentials thus providing a playground to design sustainable and cost-effective titanium-containing positive electrode materials with promising electrochemical characteristics.

---

[1] Skoltech Center for Energy Science and Technology, Skolkovo Institute of Science and Technology, 121205 Moscow, Russian Federation. [2] Department of Chemistry, Lomonosov Moscow State University, 119991 Moscow, Russian Federation. [3] Elettra Sincrotrone Trieste S.C.p.A, Area Science Park, 34012 Basovizza, Italy. ✉email: s.fedotov@skoltech.ru

The exponential growth of the number of electric vehicles as well as gradual penetration of smart grids and stationary energy storage systems into the mass market inevitably push the development of economically feasible metal-ion batteries constructed with cheap and vastly accessible raw materials[1,2]. Currently, Li and Co used in contemporary commercial batteries do not satisfy the abundance and geographical availability criteria, which defined their highly unstable price[3,4]. Striving to the affordability of grid and large-scale technologies, it was suggested to replace Li-containing materials by lower-cost Na or K counterparts preserving the same battery architecture[5–11]. As for the Co-based positive electrode (cathode) part of the battery, which is considered a central element determining energy-related properties, many Fe and Mn-based cathode materials fulfilling sustainability principles and delivering sufficient energy density and power have been sturdily pursued[12–14]. In this sense, Ti-containing electrode materials are usually disregarded because the $Ti^{(n+1)+}/Ti^{n+}$ transitions generally display an insufficiently low redox potential for cathode applications, thus being mostly employed to create negative electrode (anode) materials: $Na_2Ti_3O_7$ (0.3 V vs. $Na^+/Na$)[15], $K_2Ti_4O_9$ (~0.7 V vs. $K^+/K$)[16], $NaTi_2(PO_4)_3$ (2.1 V vs. $Na^+/Na$)[17,18], $KTi_2(PO_4)_3$ (1.8 V vs. $K^+/K$)[19] etc[20–24].

However, titanium is practically reasonable for mass applications: it is of an affordable price close to that of manganese, mined and supplied worldwide, its natural sources are rather widespread, and subsequently no political risks can be placed on. Moreover, basic Ti-containing chemicals are not hazardous or carcinogenic, readily and broadly manufactured. Once the problem with the poor electrode potential is solved, Ti-based cathode materials might turn out competitive for the mass-market.

Many works vividly demonstrated that the electrode potential of a cathode material can be tuned by adjusting the coordination environment of the $d$-metal redox center or chemical composition of the anion sublattice[25–29]. Playing with the ionicity of the Ti–X bonds (X—anionic ligand) by introducing various electron-accepting species with a so-called "inductive effect" can result in upshifting of the electrochemical potential. So far the highest electrode potential attributed to the $Ti^{4+}/Ti^{3+}$ redox couple was achieved in the titanium nitridophosphate, $Na_3Ti[(PO_3)_3N]$[30] exhibiting a voltage plateau centered at 2.7 V vs. $Na^+/Na$ (~2.95 V vs. $Li^+/Li$ or $K^+/K$) which is explained by a colossal inductive effect of the complex nitridophosphate group $[(PO_3)_3N]^{6-}$. However, because of the minor theoretical specific capacity of 72 mA h $g^{-1}$ stemming from the heavy molar mass of the $[(PO_3)_3N]^{6-}$ group, this material is not attractive enough for practical implementation[31].

A combination of a polyanion group ($XO_4^{m-}$, X = Si, P, S) and fluoride anions might present a viable strategy to further enhance the redox potential owing to the highest electronegativity of fluorine[32,33]. Indeed, fluoride-containing polyanion materials typically provide higher electrode potentials in comparison to their fluoride-free counterparts (for instance, $LiTi_2(PO_4)_3$ – 2.4 V (ref. [34]) and $LiTiPO_4F$ – 2.8 V (refs. [35,36]) vs. $Li^+/Li$, $NaTi_2(PO_4)_3$ – 2.1 V (ref. [17]) and $Na_3Ti_2(PO_4)_2F_3$ – 2.5 V (ref. [37]) vs. $Na^+/Na$, for the $Ti^{4+}/Ti^{3+}$ redox couple). Moreover, the crystal structure also plays an important role varying the redox potential attributed to the $M^{(n+1)+}/M^{n+}$ transition by up to 0.5 V or more[29,38,39].

In this paper, we report on a promising $KTiPO_4F$ fluoride phosphate as a positive electrode material for K-ion batteries (KIBs). It adopts a $KTiOPO_4$-(KTP)-type crystal structure that boosts the $Ti^{4+}/Ti^{3+}$ transition to extraordinarily high electrode potentials approaching 3.6–3.7 V vs. $K^+/K$. This earth-abundant, synthetically scalable, and thermally stable $KTiPO_4F$ material shows steady cycling at high rates thus challenging and outperforming many benchmarked potassium-ion cathode materials.

Additionally, this KTP-type structure, as was earlier shown for $KVPO_4F$[40–45], accounts for the high cycling stability and attractive rate capabilities also demonstrated by the $KTiPO_4F$ cathode material.

## Results

**Structural characterization of $KTiPO_4F$.** Stabilizing the $Ti^{3+}$ oxidation state in solids typically requires high-temperature annealing under strong reducing conditions. Alternatively, $Ti^{3+}$-containing compounds can be obtained in acidic water solutions (pH close to 1) starting from metallic Ti, which is highly soluble in HF-enriched media created by $KHF_2$. A hydrothermal treatment of such a solution at 200 °C leads to a dark-violet $KTiPO_4F$ powder.

For the as-prepared $KTiPO_4F$, no traceable known crystalline admixtures were detected by powder XRD analysis. The X-ray diffraction pattern of $KTiPO_4F$ was fully indexed on an orthorhombic lattice (S.G. #33 $Pna2_1$) with $a = 13.0020(2)$ Å, $b = 6.43420(8)$ Å, $c = 10.7636(2)$ Å, $V = 900.45(2)$ Å$^3$ (Fig. 1a). The space group choice was validated by electron diffraction: the SAED patterns of $KTiPO_4F$ (Fig. 1b) clearly demonstrate $0kl$: $k + l = 2n$, and $h0l$: $h = 2n$ reflection conditions, in agreement with the space group $Pna2_1$. Crystallographic parameters, atomic positions, atomic displacement parameters and selected interatomic distances are presented in Supplementary Tables 1–3. [100] HAADF-STEM image (Fig. 1c) confirms the refined structure. In this image, the dots correspond to the projections of the mixed Ti–P columns. The K columns are virtually invisible due to strong positional disorder violating the electron channeling.

$KTiPO_4F$ adopts a $KTiOPO_4$-(KTP)-type structure built upon helical chains of corner-shared $TiO_4F_2$ octahedra linked by $PO_4$ tetrahedra through vertexes to form a robust framework that encloses a 3D system of intersecting continuous spacious cavities accommodating $K^+$ ions (Fig. 1a, inset). The $KTiPO_4F$ unit cell is about 3.5% larger than that of $KVPO_4F$ due to a larger radius of $Ti^{3+}$ compared to $V^{3+}$ (0.67 vs. 0.64 Å, respectively, for coordination number, CN = 6).

Both K sites are disordered similar to $KVPO_4F$[40] being split into two and three K subsites for the K1 and K2 positions, respectively (Supplementary Table 2), aligned along the "helical" migration pathway characteristic to the KTP-type structures[40,45]. Such a distribution of $K^+$ ions within the voids with several energy minima anticipates their high mobility in the structure. Two symmetry inequivalent $PO_4$ tetrahedra are quite regular.

In the $TiO_4F_2$ octahedra, fluorine atoms occupy $cis$- and $trans$-positions for the Ti1 and Ti2 atoms, respectively (Fig. 1d). Average Ti–X (X = O, F) bond lengths are 2.014 Å for the Ti1 site and 2.038 Å for the Ti2 site being both characteristic to a $Ti^{3+}$ environment. The 3+ oxidation state of titanium is also validated by calculations of bond valence sums for the titanium sites (3.10(5) and 2.95(5) for Ti1 and Ti2, respectively) as well as by the EELS spectroscopy at the Ti-$L_{2,3}$ edge (Fig. 1e), EPR measurements (Fig. 1f, g), and DFT + U calculations.

The absence of crystal $H_2O$ and surface or bulk (structural) –OH groups, which could potentially replace fluorine is confirmed by the FTIR spectra of $KTiPO_4F$ revealing no corresponding absorptions in the 4000–3000 and 1600–1400 cm$^{-1}$ regions. The bands in the 1050–875 cm$^{-1}$ spectral region are characteristic to the stretching vibrations of the $PO_4$ groups, while the overlapping absorptions located at 800–600 cm$^{-1}$ match the X–Ti–X (X = O, F) stretching vibrations and O–P–O bending vibrations (Supplementary Fig. 1).

**Chemical characterization of $KTiPO_4F$.** From the TEM-EDX data the K:Ti:P ratio is equal to 0.9(1):1.13(5):0.98(8), which is

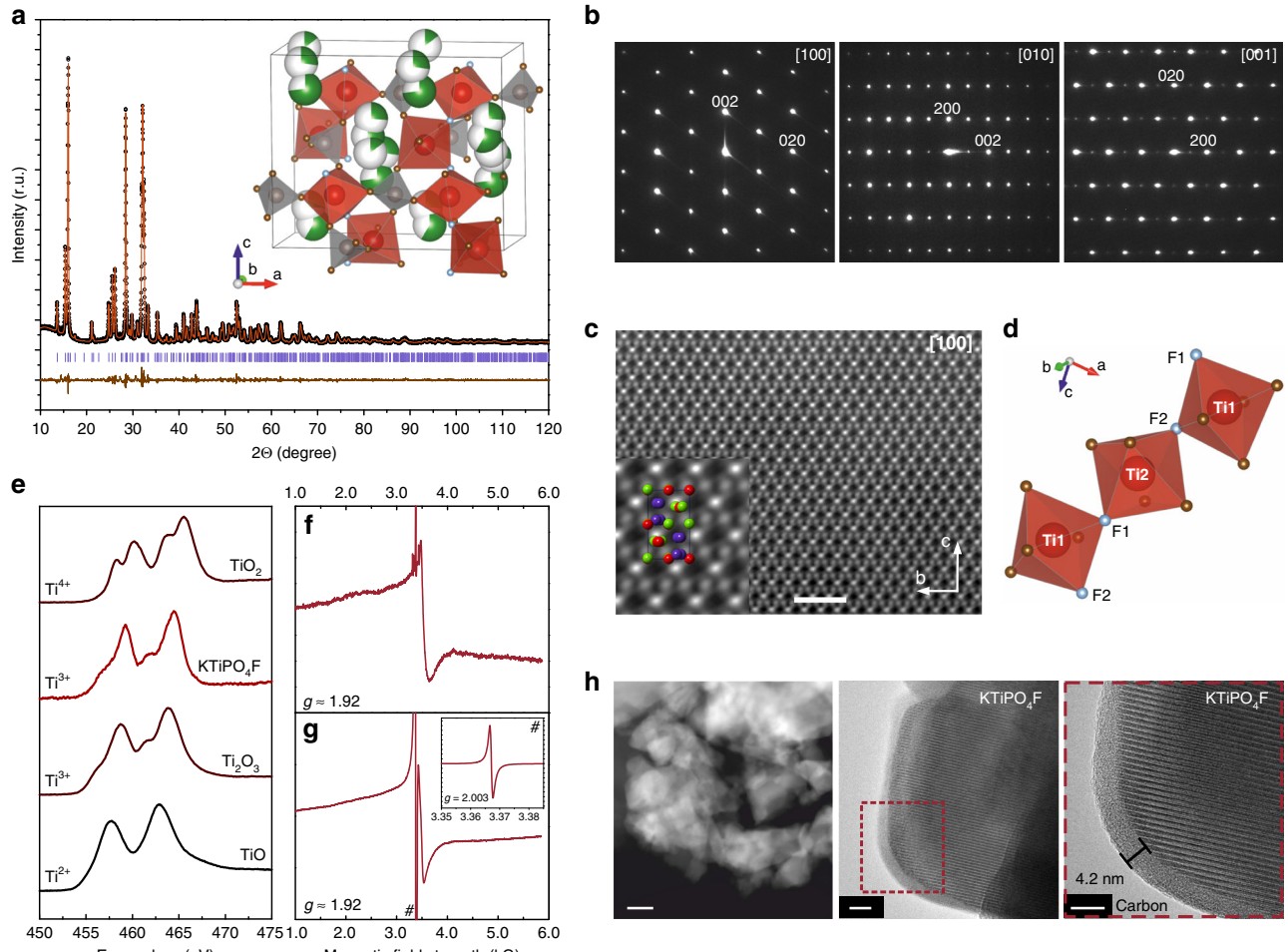

**Fig. 1 Structural and microscopic characterization of KTiPO₄F. a** Experimental, calculated and difference XRD patterns after the Rietveld refinement. Bragg reflections are denoted as violet bars. Inset: ball-polyhedral representation of the KTiPO₄F crystal structure. TiO₄F₂ octahedra are violet, PO₄ tetrahedra—orange, oxygen atoms are shown in brown, fluorine atoms—in gray. **b** Selected area electron diffraction (SAED) patterns of KTiPO₄F. **c** [100] HAADF-STEM image of KTiPO₄F. Inset: enlarged part of the HAADF-STEM image with the unit cell content overlaid. Titanium atoms are depicted in red, phosphorus is green, potassium—blue. **d** Graphical representation of fluorine atoms (gray spheres) arrangement and Ti sites coordination. **e** EELS spectrum of KTiPO₄F in the vicinity of Ti-L$_{2,3}$ edge in comparison to the reference spectra of Ti-oxides with various oxidation states of Ti. The oxide EELS spectra perfectly corroborate with those in Ref. [69]. **f** EPR spectrum of KTiPO₄F at 86 K. Broad signal with *g* factor around 1.92 indicates paramagnetic Ti³⁺ state. **g** EPR spectrum of the carbon-coated KTiPO₄F/C at 86 K. Broad signal with *g* factor around 1.92 indicates paramagnetic Ti³⁺ state. "#"—designated sharp signal with *g* = 2.003 originates from paramagnetic centers in carbon coating; inset—narrow-range EPR spectrum of the same sample showing only the sharp signal from carbon (the broad signal due to Ti³⁺ ions could not be seen in this narrow range). **h** Low-magnification HAADF-STEM image of a KTiPO₄F particle agglomerate consisting of 50–150 nm particles (left, scale bar 50 nm). High-resolution TEM images of a KTiPO₄F particle showing the carbon-coating (middle, right, scale bars 10 nm and 5 nm respectively).

close to 1:1:1 and also consistent with the ascribed formula. Slight Ti excess might originate from the residual non-dissolved metallic titanium, used as a precursor, albeit not detected by the XRD analysis.

To estimate the O:F ratio in KTiPO₄F we performed a SEM-EDX analysis by calibrating the intensities using a structurally- and chemically-related standard material with a known O:F ratio as a reference which could be a single-phase KCrPO₄F material isostructural to KTiPO₄F. Cr³⁺ is known to be a stable oxidation state, which along with the absence of OH groups guarantees a close to stoichiometric O:F ratio. The SEM-EDX data are presented in Supplementary Table 4. The O:F ratio in the reference KCrPO₄F was first estimated to be 3.97(8):1.03(8) and normalized to 4:1 yielding normalizing coefficients of 1.01 and 0.97 for O and F, respectively. In the KTiPO₄F material the averaged O:F ratio was found to be 4.06(8):0.94(8), which after normalization

gives 4.10(8):0.91(8) corresponding to the resulting formula of KTiPO$_{4.10(8)}$F$_{0.91(8)}$. Given the average value, the material seems to be slightly oxidized. However, it can be considered close to stoichiometric within the error window of determination.

Since phosphate-based materials usually display low electronic conductivity, the as-prepared KTiPO₄F powder was carbon-coated to yield the KTiPO₄F/C composite. According to the TEM data, it consists of round-shaped particles of 50 to 150 nm in diameter as shown in Fig. 1h. The particles are perfectly crystalline as seen by the high-resolution TEM imaging (Fig. 1h, right), uniformly covered with a 4–5 nm layer of carbon-coating (Fig. 1h). The Raman spectrum of the KTiPO₄F/C composite exhibits the characteristic D and G bands located at around 1350 and 1590 cm$^{-1}$, respectively (Supplementary Fig. 2). The observed peak height ratio ($I_D/I_G$) is about 1.6 indicating a sufficient disorder in the resulted carbon-coating.

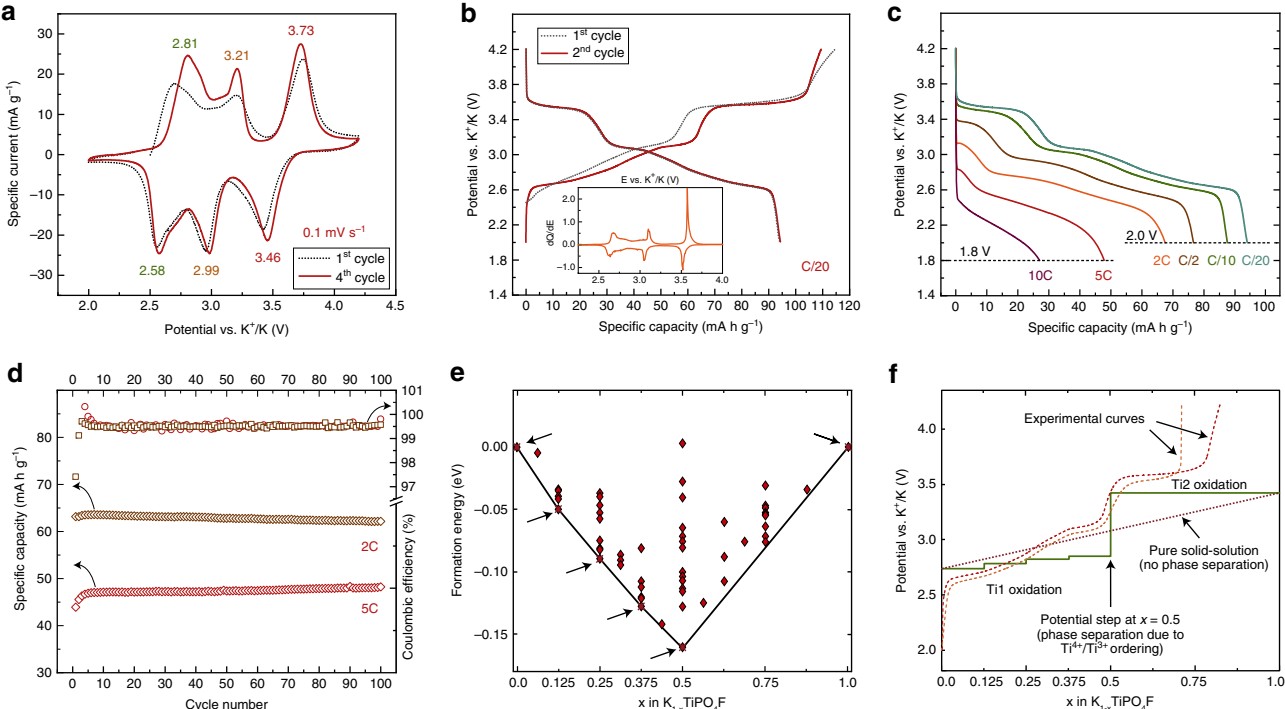

**Fig. 2 Electrochemical and computational characterization of KTiPO$_4$F. a** CVs at 100 µV s$^{-1}$ scan rate in the 2.0–4.2 V vs. K$^+$/K range. **b** First and second charge-discharge cycles at a C/20 rate. Inset: dQ/dE differential plot for the second galvanostatic cycle at C/20. **c** Galvanostatic discharge curves at rate capabilities measurements, 1C is equal to 133 mA g$^{-1}$ (C/2 charge rate was applied in all cycles starting from C/2). **d** Discharge capacities during extended cycling at 2C and 5C charge/discharge rate for 100 cycles and coulombic efficiency. **e** Formation energies calculated using DFT + U method in the K$_{1-x}$TiPO$_4$F system coupled with the clusters expansion method to predict possible ground state structures at intermediate concentrations. Crosses and arrows mark stable phases. **f** Theoretical voltage profile based on the ground state phases obtained from the convex hull (solid line), experimental charge curve (dashed line) and "gedanken" pure solid-solution voltage profile with no phase separation or charge ordering (dotted line).

**Electrochemical characterization of KTiPO$_4$F.** Electrochemical measurements revealed that the carbon-coated KTiPO$_4$F exhibits a complex charge-discharge behavior characterized by three distinct reversible processes centered at 2.7, 3.1, and 3.6 V vs. K$^+$/K as seen in the CV curves (Fig. 2a). The reversible activity at 2.7 V might be more complex including at least two overlapping processes as concluded from the asymmetry of the corresponding CV peaks. The first cycle of the CV and galvanostatic curves slightly differs from others, which might be explained by a slight oxidation of Ti$^{3+}$ to Ti$^{4+}$ in the KTiPO$_4$F/C material or SEI formation. On the galvanostatic curves (Fig. 2b) all three processes display plateau-like voltage profiles indicating a possible two-phase transformation mechanism. The discharged capacity at C/20 achieves 94 mA h g$^{-1}$. When increasing the discharge current the length of the high-voltage plateau diminishes much faster in comparison to those of the lower voltage processes, which is indicative of a kinetically more hindered K de/intercalation process (Fig. 2c). However, the material shows stable cycling at 2C and 5C (Fig. 2d) retaining almost 97% of the initial capacity at 2C and recovering a bit of extra capacity when cycled at 5C such that the specific capacity at the 100th cycle is ~7% higher than the initial one (47 mA h g$^{-1}$ vs. 44 mA h g$^{-1}$, respectively). The coulombic efficiencies for the 2C and 5C measurements are more than 99.5% (Fig. 2d).

**Theoretical calculations.** To understand the structure transformations occurring during electrochemical cycling and estimate the existence of possible stable intermediate phases, the cluster expansion method with energies based on DFT + U calculations was employed. The convex hull plot for K$_{1-x}$TiPO$_4$F is shown in Fig. 2e, which revealed several stable structures at $x = 0.125, 0.25,$

0.375, and 0.5. The phases at $x = 0.4375$ and 0.75 are very close to the convex hull which suggests that under experimental conditions these phases could be (meta)stable as well. At $x = 0.5$ a pronounced coupled K-vacancy and Ti$^{4+}$/Ti$^{3+}$ ordering is observed. At other intermediate vacancy concentrations, the ordering is weaker, which is accompanied by a high density of structures near the convex hull.

Based on the obtained ground state phases, a theoretical voltage profile was plotted (Fig. 2f). The potential is slowly increasing in three steps in the 0.0–0.5 $x$ range, which is overall in accordance with the experimental voltage profile. Some discrepancy is observed in 0–60 mA h g$^{-1}$ ($x = 0.0$–0.5), where following the experimental galvanostatic curve (Fig. 2b) the potential is increasing by 0.4 V, while the theory predicts only 0.1 V. A faster increase of the voltage in the experiment can be attributed to a solid-solution-like behavior, which is imposed on two-phase mechanisms at 2.8–3.1 V vs. K$^+$/K potentials. The formation of a solid solution can be controlled by configuration entropy and kinetic factors, which is implicitly confirmed by a high density of intermediate structures near the convex hull at $x$ lying in the [0.25:0.375] interval (Fig. 2e)[46]. According to the DFT + U calculations at $x = 0.5$, the potential upshifts by ~0.6 V, which is 0.2 V larger than in the experiment compensating underestimation of the voltage at the previous steps (Fig. 2b).

**Mechanistic understanding.** From the comparison of theoretical and experimental voltage curves, it follows that only ~0.8–0.9 K$^+$ is effectively extracted from the KTiPO$_4$F structure, afterwards the potential rapidly increases and cannot be related to the change of the Ti oxidation state that is still Ti$^{4+}$/Ti$^{3+}$. Therefore, one of the reasons is a diminishment of the K$^+$ extraction

kinetics. To get a comprehensive understanding of this phenomenon we calculated $K^+$ migration barriers. For KVPO$_4$F and RbVPO$_4$F[45], it was shown that the only relevant paths for $K^+$ migration in the KVPO$_4$F structure run along the $c$-axis and have a helical shape, with two non-equivalent hops being possible. The migration barriers for a K-vacancy in KTiPO$_4$F calculated using the NEB method do not exceed 0.2 eV, the same value is obtained for a K-vacancy in KVPO$_4$F[45]. The shape of migration pathways is depicted in Supplementary Fig. 3. However, as soon as all $K^+$ is extracted the migration barrier for $K^+$ migration increases to 0.25 eV. Due to the exponential dependence of the diffusion coefficient on migration barrier ($\exp[E_m/k_BT]$) this corresponds to the reduction of the diffusion coefficient by one order of magnitude, which will retard the extraction of the residual $K^+$ ions. The reason for the increase of migration barrier might be related to the rapid contraction of the cell volume in the 0.5–1.0 range of the K-vacancies concentration (Supplementary Table 4). Taking into account the influence of on-site electron repulsive energy U and corresponding polaron localization effects might increase this difference even further up to several orders, as we obtained for $K^+$ and $Rb^+$ migration in the VPO$_4$F framework with the same crystal structure[45].

For further examination of the phase transformations and related cell parameters changes *in operando* synchrotron powder diffraction (SXPD) and ex situ laboratory XRD experiments were carried out. The analysis of the SXPD patterns evolution during electrochemical cycling (Fig. 3a) and ex situ XRD patterns at various depths of charge/discharge (Fig. 3b, c) allowed concluding the following: i) $K^+$ intercalation in the KTiPO$_4$F is entirely reversible as the designated reflections fully recover their initial positions and intensities; ii) charge and discharge processes are symmetric, no significant discrepancy in the structural behavior between charge and discharge is observed; iii) several phase

transformations likely happen during the charge/discharge around 2.6, 3.1, and 3.6 V, as seen by abrupt changes of the cell volume with co-existence of two phases (Fig. 4b); iv) in the 2.6–3.1 V region (on charge) a solid solution mechanism is presumably characteristic for $K^+$ deintercalation as the cell volume gradually changes (Fig. 4b).

The presence of multiple two-phase intercalation mechanisms eventually explains the series of plateaus at the voltage curve centered at 2.7, 3.1, and 3.6 V and reversible oxidation/reduction peaks on the CVs and dQ/dE derivative of the charge/discharge curve (Fig. 4c).

It is interesting to speculate on the $K^+$ storage behavior during cycling. According to the SXPD data, $K^+$ deintercalation in KTiPO$_4$F starts with a two-phase transition as indicated by disappearance of the 212 (Fig. 4a, top) and 323 reflections possibly introducing C ($hkl$: $h + k = 2n$) or A ($hkl$: $k + l = 2n$) centering to the unit cell. This centering can result from switching to a higher or lower symmetry. Minimal supergroups for $Pna2_1$ with the corresponding centering conditions are $Ccm2_1$ (non-standard setting) or $Ama2$ (origin at 0 ¼ 0).

However, these space groups fail to describe adequately the observed reflections owing to extra extinction conditions characteristic to these supergroups thus requiring a monoclinic distortion leading to $Cc$ ($Cm$, $C2/c$) or $Ac$ ($Am$, $A2$) space groups. Lowering symmetry for the initial $Pna2_1$ space group from orthorhombic to monoclinic can directly result in the $Cc$ subgroup. At this point due to an insufficient resolution of the 2D detector resulting in large peak halfwidths on the SXPD patterns, making an explicit choice of the space group becomes complicated. Nevertheless, among the mentioned space groups the best fit of diffraction pattern profiles is achieved with the $Cm$ space group, which is taken for cell parameters refinement. Moreover, various $K^+$ ions orderings during extraction may

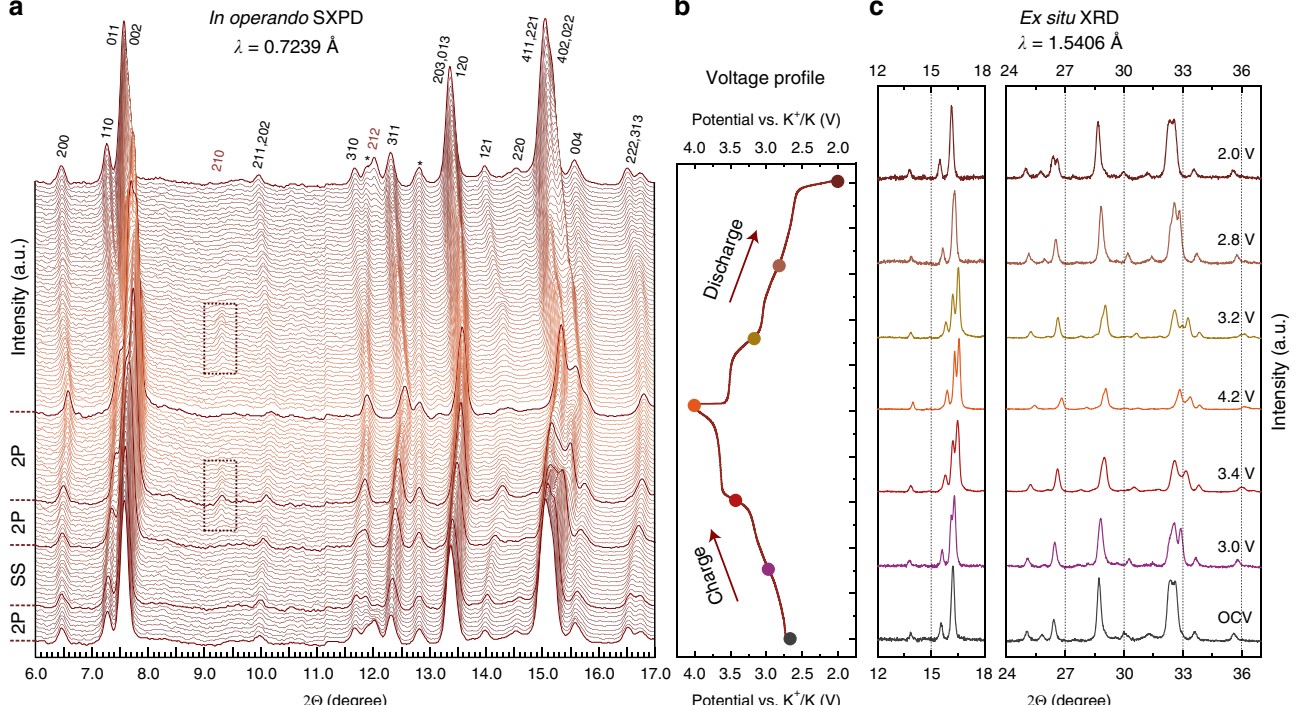

**Fig. 3 *In operando* and ex situ diffraction studies of KTiPO$_4$F. a** Diffraction patterns of the *in operando* SXPD of KTiPO$_4$F in K cell in the 6–17° 2Θ range ($\lambda = 0.7239$ Å). **b** Corresponding charge-discharge profile. Indexed reflections are denoted. The highlighted dark-red XRD patterns show tentative boundaries of the two-phase (2P) or solid solution (SS) deintercalation regimes on charge. The discharge behavior of KTiPO$_4$F can be considered symmetric. The asterisk sign (*) designates steady reflections belonging to cell components. **c** Selected regions of the ex situ XRD patterns of electrodes recovered at various potentials (charge/discharge at C/20 rate, hold for 20 h at every potential).

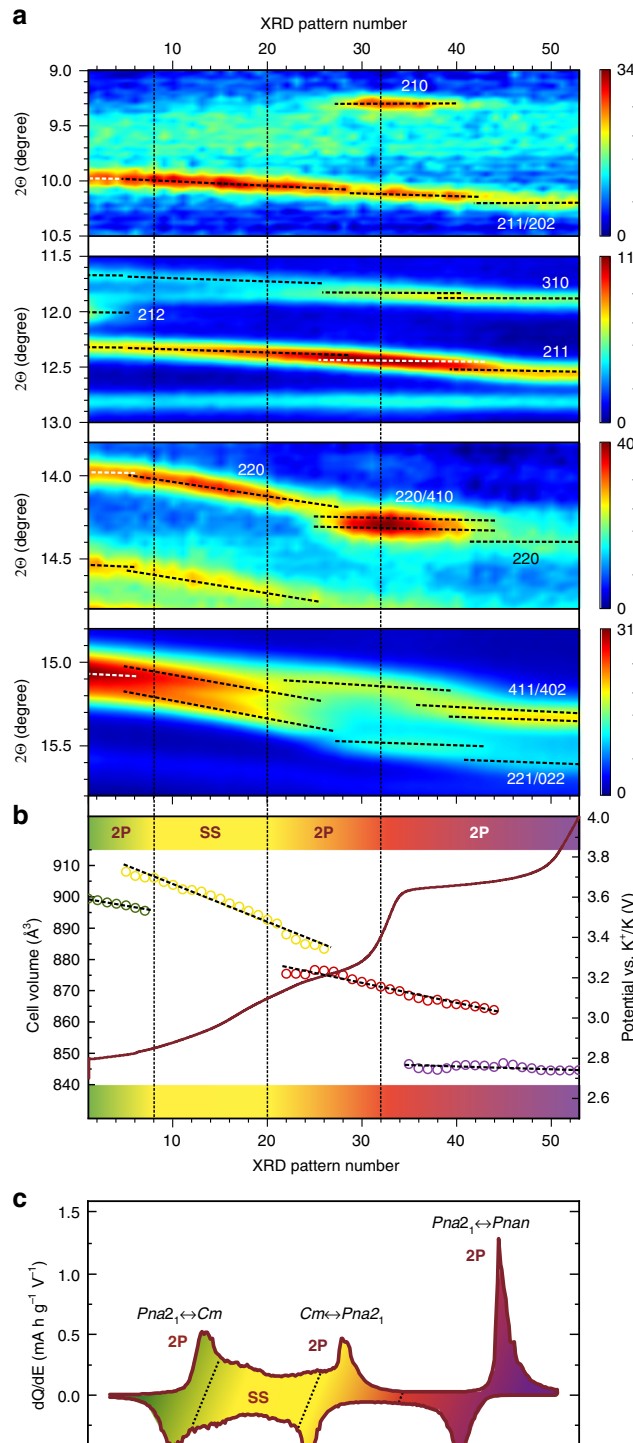

**Fig. 4 Structural evolution of KTiPO₄F during deintercalation.**
**a** Magnified regions of the *in operando* SXPD intensity map: phase transformations. White and black dashed lines designate reflections related to particular phases. 2P—two-phase region, SS—solid-solution region. Vertical dashed lines show tentative borders of 2P and SS regions. **b** Cell volume evolution on charge as refined from *in operando* SXPD data for K$_x$TiPO$_4$F with related charge galvanostatic curve. **c** dQ/dE plot showing potential ranges for solid-solution and multiple two-phase deintercalation regions.

result even into lower symmetries with larger unit cells as also evidenced by DFT for intermediate compositions (K$_{0.875}$, K$_{0.75}$, etc.) converged to monoclinic and triclinic structures. The further K$^+$ deintercalation tentatively proceeds via a solid-solution mechanism as no changes in symmetry can be found. This region is characterized by a gradual shift of peaks positions and a slope behavior of the corresponding voltage curve, and presumably ends at 3.1 V (on charge) followed by another two-phase transition. The symmetry choices for the third and fourth two-phase transition were guessed from the DFT relaxation of the corresponding structures. A short plateau centered at 3.2 V hints on the symmetry change back to *Pna2₁* with appearance of the 210 and 410 reflections at 9.3° and 14.3° 2Θ, respectively, belonging to this space group which further disappear designating a transition to the *Pnan* space group where these reflections are forbidden (in *Pnan hk0: h + k = 2n*). The crystal symmetry of the charged material well satisfies the extinction conditions of the centrosymmetric *Pnan* space group (the *hk0* reflections with *h + k ≠ 2n* are absent due to the *n*-glide perpendicular to the *c* axis). Furthermore, the crystal structures of charged electrodes of the isostructural materials (KVPO$_4$F and KFeSO$_4$F) also adopt this symmetry which was earlier demonstrated experimentally[40,47].

The crystal structure of the depotassiated K$_x$TiPO$_4$F/C material charged up to 4.2 V and held at this potential for 20 h was refined based on the ex situ XRD data on the recovered electrode (Supplementary Fig. 4, Supplementary Table 5). The polyhedral framework preserves its integrity. The cell parameters contract on charge causing a volume decrease by 6.5% to 844.80(7) Å³ (Supplementary Table 6). When compared to the cell parameters changes obtained from the DFT + U calculations, such a volume shrinkage should correspond to ~80% of the extracted K$^+$ ions if the volume change is considered linearly linked to the K$^+$ content. Indeed, the refinement of the K1 site occupancy resulted in (Supplementary Table 7) K$_{0.19(2)}$TiPO$_4$F composition being in a good accordance with the electrochemical data and supporting the DFT conclusions.

The valence evolution during the extraction/insertion of K$^+$ within the crystal structure of KTiPO$_4$F was studied by EELS for recovered electrodes charged to 3.4 V, 4.2 V and discharged to 2.0 V after full charge to 4.2 V vs. K$^+$/K. The spectra are presented in Supplementary Fig. 5a.

The position of energy offsets and maxima of the Ti-L$_{2-3}$ edges depend on the titanium valence state in KTiPO$_4$F. Moving from the initial electrode to the one charged to 4.2 V, a gradual shift towards a higher energy loss could be observed which is related to the increasing Ti oxidation state. At 2.0 V, the maxima on the EELS spectrum return to the initial positions that are characteristic for Ti$^{3+}$. Moreover, a closer look at the 3.4 V (half-charged) spectrum unveils a superposition of the Ti$^{3+}$ and Ti$^{4+}$ signals (Supplementary Fig. 5a).

Additionally, the Ti oxidation state was checked with ex situ EPR spectroscopy. The EPR spectrum of the electrode sample charged to 4.2 V contains an anisotropic signal (Supplementary Fig. 5b) manifested by residual Ti$^{3+}$ ions (according to the Rietveld refinement the chemical composition of the charged sample is K$_{0.19(2)}$TiPO$_4$F, consequently it might contain up to 20% of Ti$^{3+}$ ions). This signal can be simulated using the axial g-tensor with components $g_{xx} = g_{yy} = g_\perp = 1.916$ and $g_{zz} = g_\parallel = 1.833$ (Supplementary Fig. 5b). The fact that $g_\perp > g_\parallel$ evidences that Ti$^{3+}$ are located in distorted octahedral sites, which is in line with the structural data. Worth noting, Ti$^{4+}$ ions have a $d^0$ configuration and hence could not be detected by the EPR spectroscopy. The EPR spectrum of the electrode sample after a full charge (4.2 V)–discharge (2.0 V) cycle shows a broad signal due to paramagnetic Ti$^{3+}$ ions ($g \approx 1.92$)—quite similar to that observed in the pristine electrode material (see Supplementary

Fig. 5c). Thus, results from the ex situ EPR measurements also provide evidence in favor of a reversible $Ti^{3+}/Ti^{4+}$ redox transition during electrochemical cycling of the $KTiPO_4F$-based electrode material.

## Discussion

The synthesized $KTiPO_4F$ complements the $KMPO_4F$ (M – 3d metal) series of $KTiOPO_4$-type fluoride phosphates. Ti-based $KTiPO_4F$ represents a practically viable positive electrode material for rechargeable K-ion batteries taking several advantages over the benchmarked candidates. First, $KTiPO_4F$ can be readily prepared via a hydrothermal route, which is beneficial in controlling the morphology and particle sizes of the material after an appropriate tuning of the synthesis parameters, and can be easily scaled up. Second, $KTiPO_4F$ is based on available and affordable 3d metal, titanium, which implies no significant costs for raw materials resulting in a lower price. Third, $KTiPO_4F$ is thermally stable up to 700 °C in contrast to Prussian blue analogs (hexacyanoferrates, $K_xM^1[M^2(CN)_6]_y \cdot zH_2O$), which typically degrade over 100 °C and further evolve poisonous cyanide-based or so-called $NO_x$-based volatile products[48–52]. This thermal stability enables $KTiPO_4F$ to be carbon-coated, which in turn drastically diminishes the amount of extra carbon black added to the electrode composite. In the case of hexacyanoferrates due to their low electronic conductivity the amount of carbon black typically reaches 30%[52] and more, which correspondingly limits their practical specific energy. With lower amounts of carbon black, hexacyanoferrates show rather poor electrochemical performance. Additionally, the electrode potential of $KTiPO_4F$ ideally fits the voltage window of conventional electrolytes, which ensures its extended cycle life at various C-rates as demonstrated in this work. Moreover, the cell volume variation for $KTiPO_4F$ amounting to 6.5% is quite low for a K-ion battery electrode, especially in comparison to hexacyanoferrates with a 9–10% volume change[53].

From the crystal chemistry point of view, $KTiPO_4F$ is isostructural to $KTiOPO_4$ and $KVPO_4F$ such as no significant structural difference is observed except for the manner of the K sites splitting. A larger unit cell in the case of Ti provides more room for $K^+$ ions, which become more unevenly distributed within their crystallographic voids resulting in the above-discussed splitting of the K2 position into three partly occupied ones.

It should be particularly noted that dealing with $Ti^{3+}$, which is generally prone to oxidation, requires inert (or reducing) conditions at each step of the synthesis and electrode material preparation to preserve the oxidation state. A $Ti^{3+}$-to-$Ti^{4+}$ conversion in $KTiPO_4F$ might take place when $KTiPO_4F$ is exposed to air followed by the formation of $K_2CO_3$, which is easily observed with a time-dependent XRD of $KTiPO_4F/C$ samples in air (Supplementary Fig. 6). From the electrochemical point of view, the carbon-coated $KTiPO_4F/C$ electrode material exhibits interesting and attractive electrochemical performance and cycling stability demonstrating no capacity fading. Once its theoretical capacity is achieved, the overall energy density might reach almost 430 W h kg$^{-1}$, which makes $KTiPO_4F$ a possible alternative for many Fe-based and even Mn-based potassium battery cathodes as soon as a proper optimization of the material is done. The possibility of extraction of all K from the structure will be a subject for a separate work and published elsewhere.

It was demonstrated that in many fluoride phosphates the fluorine-to-oxygen ratio plays a significant role in governing the electrochemical performance, which was observed and thoroughly studied for many vanadium-containing fluoride phosphates. However, the influence of the F:O ratio on the electrochemical parameters is vastly complicated because both $V^{3+}/V^{4+}$ and $V^{4+}/V^{5+}$ (or $[V=O]^{2+}/[V=O]^{3+}$) redox transitions are typically electrochemically active[54]. Deviation from the stoichiometric F:O ratio thus results in changing of the electrochemical parameters (operating potential, de/intercalation mechanism, C-rates, etc.)[43,44,54–57].

In contrast, in the case of titanium, shifting the F:O ratio in $KTiPO_4F$ implies appearance of $Ti^{4+}$ which cannot be further oxidized upon depotassiation. Consequently, it means that with a higher O content the $KTiPO_{4+\delta}F_{1-\delta}$ material gradually becomes electrochemically inactive with the increasing $Ti^{4+}$ content. When the substitution of oxygen for fluorine reaches 50% in the F site ($KTiPO_4O_{0.5(1)}F_{0.5(1)}$), the specific capacity and average potential significantly drop down, the plateau at 3.6 V disappears (Supplementary Fig. 7). Approaching the $KTiPO_4O$ composition, the reversible electrochemical performance within the 2.0–4.2 vs. $K^+/K$ is fully deteriorated (Supplementary Fig. 7a). A recent paper demonstrated the applicability of $KTiOPO_4$ as an anode material for KIBs, showing the $Ti^{4+}/Ti^{3+}$ redox activity at an average potential of ~0.8 V (ref. [58]). This experimental behavior of the $KTiPO_{4+\delta}F_{1-\delta}$ is also in a good agreement with the DFT + U calculations. We considered two main intercalation steps, $K_1TiPO_{4+\delta}F_{1-\delta}/K_{0.5}TiPO_{4+\delta}F_{1-\delta}$ and $K_{0.5}TiPO_{4+\delta}F_{1-\delta}/TiPO_{4+\delta}F_{1-\delta}$ for $\delta = 0.125, 0.25$, and 0.5. Using one unit cell, the corresponding number of symmetrically non-equivalent F:O arrangements are 1, 3, and 4, respectively. For $KTiPO_4F/K_{0.5}TiPO_4F$ the influence of the F:O ratio on the electrode potential is insignificant (Supplementary Fig. 8), which is also observed on the galvanostatic curves (Supplementary Fig. 7b). However, for the $K_{0.5}TiPO_4F/K_0TiPO_4F$ transition, the potential starts upshifting rapidly as more F is replaced by O. For $\delta = 0.5$ all Ti in $K_{0.5}TiPO_{4.5}F_{0.5}$ is in the 4+ oxidation state and further extraction of $K^+$ should result in the oxygen redox, increasing the potential up to 5.5 V. Thus the 3.6 V plateau is not visible on the CV and charge/discharge curves for $KTiPO_{4.1(1)}F_{0.5(1)}$ (Supplementary Fig. 7). Therefore, the fluorine content should be kept closest to the $KTiPO_4F$ stoichiometry for realizing high-voltage performance of the material.

As it was shown by EELS and EPR measurements, the oxidation state of titanium in the material is 3+ and the BVS values for Ti are close to 3, which in accordance with the FTIR spectrum detecting no OH groups (Supplementary Fig. 1) capable of substituting fluorine and SEM-EDX analysis using a reference $KCrPO_4F$ material give us ground to assume a close-to-stoichiometric O:F ratio in $KTiPO_4F$.

A comparison of $KTiPO_4F$ and closely-related $LiTiPO_4F$ with the tavorite structure shows that in the case of $KTiPO_4F$ the average potential of $K^+$ extraction is ~0.4 V higher than that of Li removal from $LiTiPO_4F$, which is quite rare given that $K^+$ intercalation potentials are usually lower for the same chemical compositions[59]. Moreover, in $KTiPO_4F$ a stable phase at $x = 0.5$ is formed resulting in a pronounced step of the redox potential.

Speaking about the origin of the potential step at $x = 0.5$, the evident prerequisite is two non-equivalent Ti positions with different local coordinations. With respect to fluorine atoms the Ti1 position has a *cis* coordination, while the Ti2 one is *trans* coordinated (Fig. 1c), which is believed to affect the redox potential[39]. The DFT + U calculated partial density of states (pDOS) clearly illustrate that both Ti atoms form narrow 3d-bands ($t_{2g}$) centered slightly below the Fermi level, with the Ti2 3d-band being lower by 0.25 eV (Supplementary Fig. 9a, b). The peaks are separated with a small gap and show only minor hybridization with oxygen (Supplementary Fig. 9a), confirming their strong localization at Ti atoms. Such an arrangement of the peaks suggests that the extraction of the first and second halves of K are accompanied by a sequential oxidation of the Ti1 and Ti2 redox centers

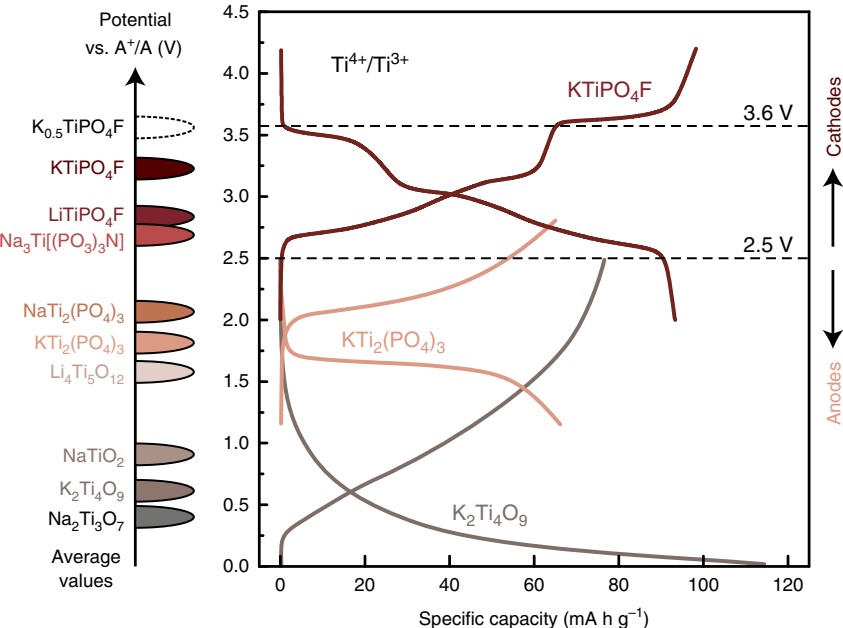

**Fig. 5 Comparison of the average operating potentials of Ti-based electrode materials and charge/discharge curves for several representatives.**
Dashed line at 2.5 V shows a tentative border between cathode (above) and anode (below) materials. Dashed line at 3.6 V designates the average potential of the $K_{0.5}TiPO_4F/TiPO_4F$ transition.

respectively. The pDOS for $K_{0.5}TiPO_4F$ (Supplementary Fig. 9c) confirms that after the extraction of 0.5 K, the Ti1 peak is emptied moving above the Fermi energy by ~2 eV, while the Ti2 peak remains occupied just below the Fermi energy. Indeed, calculations of the average intercalation potential for $KTiPO_4F/$ $K_{0.5}TiPO_4F$ with stabilization of small polaron holes exclusively on the Ti1 or Ti2 sites show that in the former case the potential is by 0.2 V lower, validating that Ti1 oxidizes first. According to the Rietveld refinement of $K_{0.46(2)}TiPO_4F$ (charged to 3.4 V, Supplementary Fig. 10, Supplementary Tables 8, 9), the BVS value for Ti1 is slightly larger than that for Ti2 (3.88(7) vs. 3.59(8)) which additionally hints at a preferential oxidation of Ti1 at $x = 0.5$. For $TiPO_4F$ both Ti1 and Ti2 peaks are emptied (Supplementary Fig. 9d).

However, the potential step at $x = 0.5$ amounts to 0.6 V (Fig. 2b, f) and cannot be comprehensively explained only by different redox potentials for the two Ti sites, therefore some other mechanism should be involved, which in our opinion has an electrostatic nature. Compared to Li-based crystal structures, due to a larger ionic radius of the $K^+$ cation, the K-based structures typically have larger volumes and, correspondingly, the $K^+$ cations are less screened by electron density. Consequently, the electrostatic interaction between $K^+$ cations is usually more pronounced than that between smaller $Li^+$ cations. For $K_{1-x}TiPO_4F$ at $0 < x < 0.5$ the minimum distance between K1 and K2 is 4.0 Å without any screening that causes strong repulsion and therefore reduction of the intercalation potential. At $x > 0.5$, after the removal of K from the K2 or K1 positions, the impact of the K-K repulsion on the free energy is eliminated due to the increase of the K–K distance to 6.4 Å and additional screening by the oxygen anions in between two $K^+$ cations. The calculation of average redox potentials from pure electrostatic energies using Bader charges confirms this conjecture, with the $K_{0.5}TiPO_4F/$ $TiPO_4F$ average potential being much larger compared to that of $KTiPO_4F/K_{0.5}TiPO_4F$. A similar effect of the potential reduction due to electrostatic repulsion was also found for the hexacyanoferrate structure, where according to DFT calculations, the potentials for Li and K are 1.91 and 0.97 V, respectively[60]. It

should be noted that in the case of a pure solid solution behavior the $K^+$ intercalation potential would increase monotonically with $x$ changing from 0 to 1 (Fig. 2f, dotted line). However, the strong ion-ion interaction causes coupled K-vacancy and $Ti^{4+}/Ti^{3+}$ ordering and change of the potential in several steps (Fig. 2f, solid line).

Returning to the reasons of the higher average potential of $KTiPO_4F$ compared to that of the $LiTiPO_4F$ tavorite structure and other Ti-based materials (Fig. 5) we attribute it to the peculiarity of the KTP crystal structure and its better energetic stabilization, since the fully deintercalated $TiPO_4F$ (KTP) and $TiPO_4F$ (tavorite) structures are very close in energy. It is a well-known fact that for the same chemical composition depending on the crystal structure the average intercalation potential can change by up to 0.5 V (ref. [29]). Moreover, previous works suggest that the increase of the average potential of the KTP structure compared to tavorite does not depend on the transition metal, providing a 0.2–0.3 V gain for vanadium as well[40,42,44]. Therefore, the KTP structure provides the largest average and absolute potentials for the given transition metal redox center.

We conclude that a combination of chemical (inductive effect) and structural (charge/vacancy ordering, electrostatics) amendments of the electrode material resulting in the extraordinary high potential for the $Ti^{4+}/Ti^{3+}$ redox activity expands the frontiers in designing attractive Ti-containing cathode materials for rechargeable batteries. This strategy may be proliferated to other $3d$ metal-based electrode materials.

## Methods

**Synthesis**. The synthesis of $KTiPO_4F$ was carried out by a hydrothermal route using a 50 ml PTFE reactor with a steel shell. Initial reagents were put in the reactor in the following weight proportion: 0.12 g of titanium powder (Ti, Sigma-Aldrich >99.9%), 0.40 g of titanyl sulfate (TiOSO₄, Sigma-Aldrich >99.9%), 2.04 g of monobasic potassium phosphate (KH₂PO₄, Sigma-Aldrich >99.9%), 0.38 g of potassium difluoride (KHF₂, Acros Organics > 99.9%), 30 g of deionized water. The reactor was maintained at 200 °C for 3 h under constant magnetic stirring and then air-cooled to room temperature. The resulting violet-colored powder was centrifuged with deionized water several times and then with acetone and dried under vacuum at room temperature. The resulting $KTiPO_4F$ powder was stored under

argon atmosphere ($pO_2 < 0.1$ ppm, $pH_2O < 0.1$ ppm). Hydrothermally prepared $KTiPO_4F$ was carbon-coated using a solution of polyvinyl cyanide $(C_3H_3N)_n$ in DMFA (Sigma-Aldrich, extra-pure) casted on the initial powder and then dried under Ar to yield the $KTiPO_4F/C$ composite by further annealing at 600 °C for 2 h (3 K min$^{-1}$ heating rate) in the highly-dried and $O_2$-purified Ar atmosphere (titanium powder was used as an oxygen absorber).

**Materials characterization**. X-ray powder diffraction (XRD) patterns were collected with a Bruker D8 Advance powder diffractometer (Cu-$K_\alpha$ radiation, 10–120° 2θ range, 0.015° 2θ step).

The energy-dispersive X-ray analysis (EDX) was performed on a scanning electron microscope (SEM) JEOL JSM-6490LV (W-cathode, operating at 30 kV) equipped with an energy-dispersive X-ray (EDX) system INCA Energy + (Oxford, Si(Li)-detector).

Thermal analysis was carried out with a TG-DSC STA-449 apparatus (Netzsch, Germany) combined with a mass spectrometer QMS 403 D Aëolos (Netzsch, Germany) under Ar or dry air flow (50 ml min$^{-1}$). The powders were heated at 10 K min$^{-1}$ rate in the 35–900 °C temperature range. The residual carbon content in the carbon-coated $KTiPO_4F/C$ was estimated to be ~6 wt.%.

The Attenuated total reflectance Fourier transform infrared (ATR-FTIR) measurements were performed with a stand-alone FTIR microscope LUMOS (Bruker) equipped with a germanium ATR crystal and liquid $N_2$ cooled MCT detector. Spectra were recorded in the 4000–600 cm$^{-1}$ range with 2 cm$^{-1}$ resolution and averaging of 64 scans. The reproducibility was checked by probing different spots of the same powder sample. All the samples were previously dried under vacuum ($p(O_2) < 10^{-2}$ atm.) at 100 °C for 24 h.

Raman spectra were obtained with a DXRxi Raman Imaging Microscope (Thermo Scientific) using 532-nm laser excitation (power at the sample plane was set to 0.5 mW) and a grating providing 2 cm$^{-1}$ spectral dispersion.

The electron paramagnetic resonance (EPR) measurements were performed at low temperature (86 ± 1 K) using a Bruker EMX-500×-band (frequency 9.44 GHz) spectrometer equipped with an ER 4119HS high-sensitivity resonator and temperature control system (Bruker). Spectra were collected with 2 mW input microwave power and a modulation amplitude of 2 G.

For the transmission electron microscopy (TEM) studies, small amount of sample was grinded with an agate mortar under ethanol. The resulting suspension was deposited onto a carbon film supported by a copper grid. Electron energy loss (EELS) spectra were collected with a Gatan GIF 2001 parallel electron spectrometer attached to an FEI Tecnai G2 transmission electron microscope operated at 200 kV. The energy resolution of the spectrometer, calculated as the full width at half-maximum height of the zero-loss peak, was 0.7 eV. The spectra were collected from an area of about 50–100 nm in diameter in a diffraction mode and far away from the zone axes to be sure that the EELS measurements are not dependent on the sample orientation. Measurements were performed under the following conditions: entrance aperture 2.0 mm, energy dispersion 0.05 eV per channel and integration time 10 s. High-resolution TEM images were obtained with a FEI Tecnai G2 transmission electron microscope. Selected area electron diffraction patterns and high angle annular dark field scanning transmission electron microscopy (HAADF-STEM) images were recorded with an aberration-corrected FEI Titan G3 transmission electron microscope at 200 kV. Ex situ EELS spectra were collected with a Titan Themis Z transmission electron microscope in a STEM mode at 200 kV accelerating voltage.

**Structural analysis**. The Rietveld refinement of the $KTiPO_4F$ structure from powder X-ray diffraction data was performed in JANA2006[61] program package with the $KVPO_4F$ structure (S.G. $Pna2_1$, #33) as an initial model[40]. The background was estimated by a set of Chebyshev polynomials, followed by fitting of the unit cell parameters and the peaks' shape with Pseudo-Voigt function. After full profile matching, scale factors, atomic coordinates and isotropic atomic displacement parameters (ADPs) were refined. At this stage, the ADPs for the positions K1 and K2 acquired three times higher values as compared to the ADPs of other atomic positions. As the EDX analysis revealed no noticeable K deficiency, static displacements were supposed to be the reason behind these high ADPs. The test refinement of the K positions in the anisotropic approximation for ADPs indicated much larger $U_{33}$ components of the thermal tensor for the both K positions. Thus, the K1 and K2 positions were split into two subsites with refined occupancies and isotropic ADPs. However, for the new K2' sub-site the isotropic ADP value did not decrease enough, consequently the procedure was reiterated yielding three K2 subsites eventually, refined with reasonable values of isotropic ADPs. For the final refinement, all ADPs were constrained to be equal for the sites occupied by the same type of element except for K.

The crystal structures were pictured using VESTA[62].

**Electrochemical measurements**. For electrochemical testing, the $KTiPO_4F$ electrodes were prepared by mixing 80 wt.% of the $KTiPO_4F/C$ active material, 10 wt.% acetylene black (carbon Super-C) and 10 wt.% polyvinylidene fluoride (PVDF). N-Methyl pyrrolidinone (NMP) was added to dissolve PVDF. The resulting slurry was cast on the aluminum foil by Doctor Blade technique. All the above-mentioned procedures were carried out in the Ar-filled glove box (MBraun, $pO_2 < 0.1$ ppm,

$pH_2O < 0.1$ ppm). The electrodes were dried in vacuum at 80 °C for 3 h to remove residual NMP. The material composite loading after drying was 5–6 mg cm$^{-2}$. The electrochemical properties of $KTiPO_4F$ were evaluated using Swagelok-type two-electrode cells with Al and Cu current collectors, $KPF_6$ electrolyte solution (0.5 M $KPF_6$ (Sigma-Aldrich, 99.5%) in EC:DEC = 1:1 vol. Sigma-Aldrich anhydrous), glass-fiber separators and potassium metal (Sigma-Aldrich) negative electrode. The galvanostatic and cyclic voltammetry measurements were carried out at room temperature using a Biologic VMP3 potentiostat.

**Ex situ and in operando X-ray diffraction**. Composite electrodes for ex situ and in operando X-ray diffraction analyses were prepared in accordance with the above-mentioned scheme (the material loading of 8–10 mg cm$^{-2}$). Ex situ diffraction data were performed on a BRUKER D8 Advance diffractometer (reflection mode, Ar-filled sealed holder, LYNXEYE detector, Cu-$K_\alpha$ radiation, at a scanning step of 0.02°). In operando synchrotron XRD experiments were performed at the MCX beamline (Elettra Synchrotron light source in Trieste, Italy)[63] at a wavelength of λ = 0.7293 Å in a transmission geometry using the marCCD-SX-165 2D detector. Modified CR-2032 coin-type cells with Kapton®-glued windows were fabricated and assembled with the $KTiPO_4F$ cathode material, glass-fiber separator and metallic K anode. 2D powder diffraction patterns were processed with GSAS-II software (build 4195)[64], integrated 1D diffraction patterns were analyzed with JANA-2006 program. Si powder (SRM640c from NIST) was used as a standard. The data were collected every 5 min under a galvanostatic cycling with potential limitation (GCPL) experiment at C/7 charge and C/14 discharge rates in the potential range of 2.0–4.2 V vs. $K^+/K$ using a Biologic SP-150 potentiostat.

**DFT calculations**. All density functional theory (DFT) calculations were performed in the VASP program[65] using generalized gradient approximation (GGA) to exchange-correlation functional and standard PAW PBE potentials[66] with a minimum number of valence electrons. To take into account the strongly correlated character of the $d$-electrons of Ti, a Hubbard-like correction is added within the Dudarev scheme[67] and the U value of 3.2 eV, which was chosen in such a way to reproduce the experimentally observed average intercalation potential of $K^+$. Gaussian smearing with a smearing width of 0.1 eV was used for Brillouin-zone integrations. All calculations were performed with spin polarization with the ferromagnetic ground state. The energy cut-off was fixed at 400 eV; the Gamma-centered k-point was $4 \times 2 \times 2$ for the unit cell with 64 atoms. To eliminate Pulay errors the lattice optimization (ISIF = 4) was performed at the constant volume for several contracted and expanded cells (7 points).

The $K^+$ migration barriers were determined using the nudged elastic band (NEB) method and GGA PBE without U as implemented in VASP using the $2 \times 1 \times 1$ supercell. The method allows finding the minimum energy path, which includes several intermediate configurations (images) between initial and final states (five intermediate images were used).

The cluster expansion method, such as implemented in the ATAT code, was used to study phase stabilities[68]. The root mean square error of the energies from the resulted cluster expansion Hamiltonian with respect to DFT energies was estimated to be less than 0.015 eV per $K^+$ ion.

The optimization of atomic positions was performed using a quasi-Newton algorithm until the maximum force permitted for any vector component was less than 0.2 eV Å$^{-1}$ for NEB calculations and 0.05 eV Å$^{-1}$ for all other calculations. The computational setup including errors due to the periodic boundary conditions has been estimated to provide the precision of 0.1 eV for migration barriers.

## Data availability

All relevant data that support the findings of this study are presented in the manuscript and supporting information. Source data are available from the corresponding author upon reasonable request.

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

## Acknowledgements

This work is supported by Russian Science Foundation (grant 18-73-00313). The authors thank Dr. O. A. Drozhzhin for the fruitful discussions on *in operando* experiments and Prof. A. Kh. Vorobiev and Dr. I. Sorokin for the assistance with the EPR measurements. The authors are grateful to Skolkovo Institute of Science and Technology (#2016-1/NGP) and Moscow State University Program of Development up to 2020. The AICF of Skoltech is acknowledged for granting access to the TEM facilities. The authors also acknowledge the CERIC-ERIC Consortium for the access to experimental facilities at MCX.

## Author contributions

S.S.F. designed the study, carried out crystallographic analysis, ex situ X-ray powder diffraction and composed the manuscript, N.D.L. performed synthesis and electrochemical characterization, D.A.A. performed DFT calculations, A.V.M. and A.M.A. performed HAADF-STEM, SAED, energy-dispersive X-ray and EELS measurements, S.V.R. collected and interpreted FTIR, Raman and EPR spectra, M.G. and J.R.P assisted with SXPD *in operando* experiments and processed the obtained data, S.S.F., K.J.S, A.M.A., and E.V.A. supervised the study, planned the experiments and analyzed the data.

## Competing interests

The authors declare no competing interests.
