## [Peer Review File · Nature Communications]

Reviewers' comments:

Reviewer #1 (Remarks to the Author):

It is interesting to solve the problems in front of the development of potassium ion batteries (PIBs), especially to develop high-performance cathode materials to boost the fast migration of potassium ion batteries and to understanding the mechanisms behind.

In this work, a novel polyanion cathode materials KTiPO_4F is developed for PIBs with a relatively higher capacity. It is also interesting to see the introduction of $\text{Ti}^{3+}/\text{Ti}^{4+}$ redox pair for energy storage. The crystal structure and electrochemical performances were investigated. This work can be a good reference for other researchers who are working on cathode materials for potassium ion batteries. However, there are still some important issues should be clarified before accepted for publication with a major revision.

1. The valance evolution during the extraction/insertion of potassium ion within the crystal structure of KTiPO_4F is recommended to be investigated by in situ or ex-situ measurement for further understanding the energy storage mechanism.

2. It can be seen from the CV and charge/discharge profile that there are three main oxidation processes and caused two electrochemical energy storage mechanisms, namely, two-phase mechanism (>3.5 V) and solid solution mechanism ($2\sim 3.5$ V). However, this observation is contradict with the claim given in Figure 2E. The operando XRD of KTiPO_4F in PIBs present in Figure 3a is not clear enough to see the new phase growth during the two phase evolution during the charge/discharge process above 3.5 V. Please clarify the concerns properly.

3. Please check the whole manuscript to avoid any similar mistakes found from line 335 in Page 15. The claimed applied titanium sources is titanyl phosphate, but the provided chemical formula was TiOSO_4 .

Reviewer #2 (Remarks to the Author):

High $\text{Ti}^{4+}/\text{Ti}^{3+}$ redox potential realized in phosphate fluoride system is very astonishing to see its operation at over 2.5 V. The authors also know how to produce trivalent Ti^{3+} that is formed in a strong acidic medium.

The material was well characterized by many methods and the authors have confirmed insertion of the large K^+ into the host structure.

Can the authors provide moisture-sensitivity of the KTiPO_4F ?

What are the used carbon sources?

Line 399. N should be italic.

Reviewer #3 (Remarks to the Author):

The work by Stanislav S. Fedotov et al. reported the synthesis and characterization of KTiPO_4F , which showed the possibility of delivering a high voltage around 3.6 V (vs. K^+/K) by utilizing the $\text{Ti}^{3+}/\text{Ti}^{4+}$ redox. A cumulative inductive effect from the phosphate and fluoride anions and the K^+ /vacancy ordering were proposed for the high voltage character. It's encouraging, although not fully out of expectation, to observe that the key electrochemical parameter of the charge/discharge plateau could

be efficiently modulated through materials design to ensure a favorable battery performance, which is helpful to envision new protocols for materials innovation for KIBs. Despite of the large amount of characterizations described in the manuscript, the manuscript shows shortcomings in provide convincing evidence to support the key conclusions claimed by the authors. Therefore I'm not able to suggest an acceptance of this work for its publication in Nat Commun. in consideration of the high standard enforced by the journal. Below are some detailed comments I have during reading the manuscript.

1. Electrochemical characterization is needed to confirm the activity of Ti^{3+}/Ti^{4+} redox. For example, EELS and EPR data at different charge states, particularly charged to 4.2 V or discharged to 2 V, should be provided to validate the electrochemical transition of Ti^{3+}/Ti^{4+} redox. The Ti contribution to the high voltage character seems to be the key argument of this work. Unfortunately, no data related to the charge compensation mechanism is provided, which makes the claim highly suspicious.
2. In fluorophosphates, fluorine: oxygen ratio seems to play a critical role in determining the electrochemical properties. However, the authors failed to provide experimental evidence and necessary discussion about the key factor. Hence, more systematic research may be required for this work, including the influence of fluorine on the operating voltage, galvanostatic curves and reversible capacity.
3. The author considered the charge/discharge during 2.0-3.4V as solid-solution-like process, despite of clearly two oxidation peaks at 2.81 and 3.21 V for charge and two reduction peaks at 2.58 and 2.99 V for discharge, which demonstrate multiple phase transitions during the reaction. A close look in the operando XRD data did show the emergence of tiny peaks during the disappearance of (212) peak of the pristine sample. For example, some tiny ones at 29.7° can be observed when charged to 3.0V. A detailed discussion is needed to facilitate the understanding of structural evolution during the electrochemical reaction.
4. For the rate capability of the tested sample as shown in Figure 4c, the authors used different voltage windows, is there a special reason for such an operation? How about the Coulombic efficiency of the sample? The authors should compare at least the first 2 charge-discharge curves at 0.1 C of the samples, which are critical to understand the structural stability and transition of the electrode materials during cycling.

Reviewer #1:

Comments to the author:

It is interesting to solve the problems in front of the development of potassium ion batteries (PIBs), especially to develop high-performance cathode materials to boost the fast migration of potassium ion batteries and to understanding the mechanisms behind.

In this work, a novel polyanion cathode materials KTiPO_4F is developed for PIBs with a relatively higher capacity. It is also interesting to see the introduction of $\text{Ti}^{3+}/\text{Ti}^{4+}$ redox pair for energy storage. The crystal structure and electrochemical performances were investigated. This work can be a good reference for other researchers who are working on cathode materials for potassium ion batteries. However, there are still some important issues should be clarified before accepted for publication with a major revision.

Question 1

The valence evolution during the extraction/insertion of potassium ion within the crystal structure of KTiPO_4F is recommended to be investigated by in situ or ex-situ measurement for further understanding the energy storage mechanism.

Response:

The valence evolution was studied by EELS for recovered electrodes charged to 3.4 V, 4.2 V and discharged to 2.0V after full charge to 4.2 V vs. K^+/K . The spectra are presented in Figure R1.

Figure R1. EELS spectra of recovered electrode materials. Ti_2O_3 and TiO_2 EELS spectra are given for reference.

The position of energy offsets and maxima of the Ti-L2-3 edges depend on the titanium valence state in KTiPO_4F . Moving from the initial electrode to one charged to 4.2 V a gradual shift towards higher energy loss can be observed which is related to the increasing Ti oxidation state. At 2.0 V the maxima on the EELS spectrum return to the initial positions that are characteristic for Ti^{3+} . Ti_2O_3 and TiO_2 spectra are given for reference. Moreover, a closer look at the 3.4 V (half-charged) spectrum unveils a superposition of Ti^{3+} and Ti^{4+} signals can be seen.

Additionally, the Ti oxidation state was checked with *ex situ* EPR spectroscopy. The EPR spectrum of the electrode charged to 4.2V contains an anisotropic signal (Fig. R2, A) manifested by residual Ti^{3+} ions (according to the Rietveld refinement the chemical composition of the charged sample is $\text{K}_{0.18(2)}\text{TiPO}_4\text{F}$, consequently it might contain up to 20% of Ti^{3+} ions). This signal can be simulated using the axial g -tensor with components $g_{xx} = g_{yy} = g_{\perp} = 1.916$ and $g_{zz} = g_{\parallel} = 1.833$ (Figure R2, A). The fact that $g_{\perp} > g_{\parallel}$ evidences that Ti^{3+} are located in distorted octahedral sites, which is in line with the structural data. Worth noting, Ti^{4+} ions have a d^0 configuration and hence could not be detected by the EPR spectroscopy. The EPR spectrum of the electrode after a charge(4.2V)-discharge(2.0V) cycle shows a broad signal due to paramagnetic Ti^{3+} ions ($g \approx 1.92$) – quite similar to that observed in the pristine electrode material (see Fig. R2, B). Thus, results from the *ex situ* EPR measurements also evidence a reversible $\text{Ti}^{3+}/\text{Ti}^{4+}$ redox transition during electrochemical cycling of the KTiPO_4F -based electrode material.

Figure R2. A) Experimental and simulated EPR spectra for the KTiPO_4F electrode charged to 4.2 V. B) Experimental EPR spectra of the initial and discharged to 2.0 V (after charge to 4.2 V) KTiPO_4F electrodes.

Moreover, Rietveld refinement of the crystal structures of partially depotassiated $\text{K}_{1-x}\text{TiPO}_4\text{F}$ revealed increasing BVS values for the Ti sites on oxidation. In the pristine KTiPO_4F BVS for the Ti1 and Ti2 atoms are 3.10(5) and 2.95(5) respectively, in the $\text{K}_{0.46}\text{TiPO}_4\text{F}$ charged to 3.4 V 3.88(7) and 3.59(8) and for the $\text{K}_{0.18(2)}\text{TiPO}_4\text{F}$ charged material 4.3(2) for both sites.

Both Figure R1 and R2 are included into the manuscript as a part of supporting information.

The following text was added to the manuscript:

“The valence evolution during the extraction/insertion of K^+ within the crystal structure of KTiPO_4F was studied by EELS for recovered electrodes charged to 3.4 V, 4.2 V and discharged to 2.0V after full charge to 4.2 V vs. K^+/K . The spectra are presented in Figure S5.

The position of energy offsets and maxima of the Ti-L_{2-3} edges depend on the titanium valence state in KTiPO_4F . Moving from the initial electrode to one charged to 4.2 a gradual shift towards higher energy loss can be observed which is related to the increasing Ti oxidation state. At 2.0 V the maxima

on the EELS spectrum return to the initial positions that are characteristic for Ti^{3+} . Moreover, a closer look at the 3.4 V (half-charged) spectrum unveils a superposition of Ti^{3+} and Ti^{4+} signals (Figure S5). Additionally, the Ti oxidation state was checked with *ex-situ* EPR spectroscopy. The EPR spectrum of the electrode sample charged to 4.2V contains an anisotropic signal (Fig. S5, A) manifested by residual Ti^{3+} ions (according to the Rietveld refinement the chemical composition of the charged sample is $\text{K}_{0.18(2)}\text{TiPO}_4\text{F}$, consequently it might contain up to 20% of Ti^{3+} ions). This signal can be simulated using the axial g-tensor with components $g_{xx} = g_{yy} = g_{\perp} = 1.916$ and $g_{zz} = g_{\parallel} = 1.833$ (Figure S5, A). The fact that $g_{\perp} > g_{\parallel}$ evidences that Ti^{3+} are located in distorted octahedral sites, which is in line with the structural data. Worth noting, Ti^{4+} ions have a d^0 configuration and hence could not be detected by the EPR spectroscopy. The EPR spectrum of the electrode sample after a full charge(4.2V)-discharge(2.0V) cycle shows a broad signal due to paramagnetic Ti^{3+} ions ($g \approx 1.92$) – quite similar to that observed in the pristine electrode material (see Figure S5, B). Thus, results from the *ex situ* EPR measurements also provide evidence in favor of a reversible $\text{Ti}^{3+}/\text{Ti}^{4+}$ redox transition during electrochemical cycling of the KTiPO_4F -based electrode material.”

Question 2

It can be seen from the CV and charge/discharge profile that there are three main oxidation process and caused two electrochemical energy storage mechanism, namely, two-phase mechanism (>3.5 V) and solid solution mechanism (2~3.5 V). However, this observation is contradict with the claim given in Figure 2E. The operando XRD of KTiPO_4F in PIBs present in Figure 3a is not clear enough to see the new phase growth during the two-phase evolution during the charge/discharge process above 3.5 V. Please clarify the concerns properly.

Response:

To elucidate the concerns regarding the phase transformation behavior of KTiPO_4F we performed a deeper technical analysis of the diffraction pattern evolution during operando experiment (Figure R3). Indeed, the phase transformation profile of KTiPO_4F during de/intercalation might be considered complicated since it contains several two-phase and solid-solution regions. The presence of two-phase transitions is in line with the cell parameters change when the cell volume abruptly changes. At the same time, multiple reversible oxidation/reduction peaks are observed on the CV curves and dQ/dE derivative of the charge/discharge curve. In the 2.8 – 3.1 V region on charge a solid

solution mechanism is presumably characteristic for K^+ deintercalation as the cell volume gradually decreases.

Figure R3. K_xTiPO_4F cell volume evolution on charge refined from operando XRD data. White and black dotted lines designate reflections related to particular phases. 2P – two-phase region, SS – solid-solution region. V – cell volume of the initial material. The corresponding charge galvanostatic curve is shown in red.

The following text along with the Figure R3 (now Figure 4) have been added to the manuscript:

“several phase transformations likely happen during the charge/discharge around as seen by abrupt changes of the cell volume with co-existence of two phases (Figure 4, top); iv) in the 2.8 – 3.1 V region (on charge) a solid solution mechanism is presumably characteristic for K^+ deintercalation as the cell

volume gradually decreases (Figure 4, bottom). The presence of multiple two-phase intercalation mechanisms eventually explains the series of plateaus at the voltage curve centered at 2.7, 3.1 and 3.6 V and reversible oxidation/reduction peaks on the CVs and dQ/dE derivative of the charge/discharge curve (Figure 2A, B – inset).”

Question 3

Please check the whole manuscript to avoid any similar mistakes found from line 335 in Page 15. The claimed applied titanium sources is titanyl phosphate, but the provided chemical formula was TiOSO_4 .

Response:

Corrected. “phosphate” appeared to be a typo. For sure, it should be titanyl **sulfate**, TiOSO_4 . The manuscript was double-checked to exclude similar errors. Thank you.

Reviewer #2:

Comments to the author:

High $\text{Ti}^{4+/3+}$ redox potential realized in phosphate fluoride system is very astonishing to see its operation at over 2.5 V. The authors also know how to produce trivalent Ti^{3+} that is formed in a strong acidic medium.

The material was well characterized by many methods and the authors have confirmed insertion of the large K^+ into the host structure.

Question 1

Can the authors provide moisture-sensitivity of the KTiPO_4F ?

Response:

Ti^{3+} generally displays poor stability when exposed to moisture or air. That is why dealing with KTiPO_4F requires inert (or reducing) conditions at each step to preserve its chemical composition and Ti^{3+} oxidation state. A Ti^{3+} -to- Ti^{4+} conversion in KTiPO_4F can take place according to the following possible reactions:

$\text{KTi}^{3+}\text{PO}_4\text{F} + y/2\text{H}_2\text{O} + y/4\text{O}_2 \text{ (from air)} \rightarrow \text{K}_{1-y}\text{Ti}^{(3+y)+}\text{PO}_4\text{F} + y\text{KOH}$ with further catching of CO_2 from air by KOH to form K_2CO_3 .

Both presumable reactions lead to formation of Ti^{4+} -enriched materials showing reduced cell parameters. In the second case formation of K_2CO_3 can be easily observed with a time-dependent XRD of samples exposed to air:

Figure R4. Time-resolved XRD patterns of the $\text{KTiPO}_4\text{F}/\text{C}$ electrode material exposed to air. Insets: the shift of the peaks confirms unit cell shrinkage due to Ti oxidation and K^+ loss. Asterisk (*) designates the peak of K_2CO_3 .

In summary, the overall degradation reaction can be written as follows:

As seen, water may not even take part in the degradation mechanism of $\text{KTi}^{3+}\text{PO}_4\text{F}$ as evidenced by the KTiPO_4F stability under the synthesis conditions (highly acidic medium, pH ~1-2, reducing conditions created by hydrogen from dissolving of metallic Ti) as well as in pure degassed (deoxygenated) water.

A corresponding paragraph has been added to the Discussion section of the manuscript:

“It should be particularly noted that dealing with Ti^{3+} , which is generally prone to oxidation, requires inert (or reducing) conditions at each step of synthesis and electrode material preparation to preserve the oxidation state. A Ti^{3+} -to- Ti^{4+} conversion in KTiPO_4F might take place when KTiPO_4F is exposed to air following with a formation of K_2CO_3 , which is easily observed with a time-dependent XRD of $\text{KTiPO}_4\text{F}/\text{C}$ samples exposed to air (Figure S8).”

Figure R4 and additional comments are included in the supporting materials.

Question 2

What is the used carbon sources?

Response:

Since Ti^{3+} is sensitive to oxidation especially at elevated temperatures, for carbon-coating we used oxygen-free carbon sources. Conventional oxygen-containing organic additives like glucose, sucrose, ascorbic acid, etc. do not work. In this sense, polyacrylonitrile (PAN) casted on the initial powder from a DMFA solution (under Ar atmosphere) turned out to be an ideal recipe to successfully perform carbon-coating of KTiPO_4F .

The following part is added to the experimental section:

“Hydrothermally prepared KTiPO_4F was carbon-coated using a solution of polyacrylonitrile (PAN) in DMFA casted on the initial powder and then dried under Ar to yield $\text{KTiPO}_4\text{F}/\text{C}$ composite by further annealing at 600°C for 2 hours (3K/min heating rate) in the O_2 -purified Ar atmosphere (titanium powder was used as an oxygen absorber).”

Question 3

Line 399. N should be italic.

Response:

Corrected. *N*-Methyl pyrrolidinone (*N* is italic). Thank you.

Reviewer #3:

Comments to the author:

The work by Stanislav S. Fedotov et al. reported the synthesis and characterization of KTiPO_4F , which showed the possibility of delivering a high voltage around 3.6 V (vs. K^+/K) by utilizing the $\text{Ti}^{3+}/\text{Ti}^{4+}$ / redox. A cumulative inductive effect from the phosphate and fluoride anions and the K/vacancy ordering were proposed for the high voltage character. It's encouraging, although not fully out of expectation, to observe that the key electrochemical parameter of the charge/discharge plateau could be efficiently modulated through materials design to ensure a favorable battery performance, which is helpful to envision new protocols for materials innovation for KIBs. Despite of the large amount of characterizations described in the manuscript, the manuscript shows shortcomings in provide convincing evidence to support the key conclusions claimed by the authors. Therefore, I'm not able to suggest an acceptance of this work for its publication in Nat Commun. in consideration of the high standard enforced by the journal. Below are some detailed comments I have during reading the manuscript.

Question 1

Electrochemical characterization is needed to confirm the activity of $\text{Ti}^{3+}/\text{Ti}^{4+}$ redox. For example, EELS and EPR data at different charge states, particularly charged to 4.2 V or discharged to 2 V, should be provided to validate the electrochemical transition of $\text{Ti}^{3+}/\text{Ti}^{4+}$ redox. The Ti contribution to the high voltage character seems to be the key argument of this work. Unfortunately, no data related to the charge compensation mechanism is provided, which makes the claim highly suspicious.

Response:

We confirmed that the charge compensation mechanism is based on the reversible activity of Ti^{3+}/Ti^{4+} redox transition, which was validated by ex-situ EELS and EPR on the recovered electrode materials. Please see the response to Question 1 of Reviewer 1.

Question 2

In fluorophosphates, fluorine: oxygen ratio seems to play a critical role in determining the electrochemical properties. However, the authors failed to provide experimental evidence and necessary discussion about the key factor. Hence, more systematic research may be required for this work, including the influence of fluorine on the operating voltage, galvanostatic curves and reversible capacity.

Response:

Indeed, the fluorine-to-oxygen ratio (F:O) in fluoride-phosphates plays a significant role in governing their electrochemical performance, which was observed and thoroughly studied for many vanadium-containing fluoride-phosphates, where the situation is far more complicated because both V^{3+}/V^{4+} and V^{4+}/V^{5+} (or $[V=O]^{2+}/[V=O]^{3+}$) redox transitions are typically electrochemically active [1]. Deviation of the stoichiometric F:O ratio thus results in changing of electrochemical parameters (operating potential, de/intercalation mechanism, C-rates etc.) [1]–[6].

On the contrary, in case of titanium, shifting F:O ratio in $KTiPO_4F$ implies appearance of Ti^{4+} which cannot be further oxidized upon depotassiation. Consequently, it means that in practice with higher O content the $KTiPO_4F$ material gradually becomes electrochemically inactive with Ti^{4+} content increasing as shown in Figure R5. Approaching the $KTiPO_4O$ composition, reversible electrochemical performance within the 2.0 – 4.2 vs K^+/K potential region fully disappears. A recent paper demonstrated the applicability of $KTiOPO_4$ as an anode material for KIBs, showing the Ti^{4+}/Ti^{3+} redox activity at an average potential of ~ 0.8 V [7].

Figure R5. CV curves on KTiPO_4F and KTiPO_4O .

As it was shown by EELS and EPR measurements, the oxidation state of titanium in the material is $3+$ and the BVS values for Ti are close to 3, which in accordance with the FTIR spectrum detecting no OH-groups capable to substitute fluorine give us ground to assume a close to stoichiometric O:F ratio in KTiPO_4F .

Additionally, to estimate the O:F ratio in KTiPO_4F we performed chemical analysis using SEM-EDX. Though absorption and Auger-electron emission processes prevent independent determination of the absolute content of oxygen or fluorine, in fact the F:O ratio value can be reliably estimated if the intensities are calibrated using a structurally- and chemically-related reference material with a known F:O ratio.

For this purpose, we synthesized a single-phase KCrPO_4F material isostructural to KTiPO_4F . Cr^{3+} is known to be a stable oxidation state, which along with the absence of OH-groups (confirmed by FTIR and TG-DSC+MS) guarantees near stoichiometric O:F ratio. The SEM-EDX data are presented in Table R1.

Table R1. SEM-EDX data and processing of O:F ratio* in KCrPO₄F (reference) and KTiPO₄F

Point	KCrPO ₄ F				KTiPO ₄ F			
	O, at. %	F, at. %	O per f.u.	F, per f.u.	O, at. %	F, at. %	O, per f.u.	F, per f.u.
1	48.52	12.02	4.01	0.99	39.23	9.69	4.01	0.99
2	36.65	9.08	4.01	0.99	28.19	5.83	4.14	0.86
3	45.75	12.14	3.95	1.05	44.09	10.33	4.05	0.95
4	46.87	13.4	3.89	1.11	27.48	5.52	4.16	0.84
5	50.33	14.95	3.85	1.15	38.70	9.00	4.06	0.94
6	47.13	12.56	3.95	1.05	31.99	6.64	4.14	0.86
7	44.13	10.53	4.04	0.96	43.69	11.40	3.97	1.03
8	40.9	10.27	4.00	1.00	30.86	7.38	4.04	0.96
9	42.04	9.92	4.05	0.95	47.18	12.72	3.94	1.06
10	36.31	8.19	4.08	0.92	47.39	11.10	4.05	0.95
11	37.37	8.47	4.08	0.92	47.31	12.75	3.94	1.06
12	50.59	14.86	3.86	1.14	29.53	6.03	4.15	0.85
13	46.96	12.83	3.93	1.07	46.57	10.76	4.06	0.94
14	48.19	14.15	3.87	1.13	34.63	7.86	4.08	0.92
15	49.94	13.12	3.96	1.04	40.44	8.30	4.15	0.85
Average			3.97	1.03			4.06	0.94
Std			0.08	0.08			0.08	0.08
Norm. average							4.10	0.91

* – ratio normalized per formula unit given that O + F = 5

The O:F ratio in the reference KCrPO₄F was first estimated to be 3.97(8):1.03(8) and normalized to 4:1 yielding normalizing coefficients of 1.01 and 0.97 for O and F respectively. In the KTiPO₄F material the averaged O:F ratio was found to be 4.06(8):0.94(8), which after normalization gives 4.10(8):0.91(8) corresponding to the resulting formula of KTiPO_{4.10(8)}F_{0.91(8)}. Given the average value, the material seems to be slightly oxidized. However, it can be considered close to stoichiometric within the error window.

A paragraph was added to the manuscript:

“To estimate the O:F ratio in KTiPO₄F we performed a SEM-EDX analysis by calibrating the intensities using a structurally- and chemically-related standard material with a known O:F ratio as a reference

which could be a single-phase KCrPO_4F material isostructural to KTiPO_4F . Cr^{3+} is known to be a stable oxidation state, which along with the absence of OH-groups guarantees a close to stoichiometric O:F ratio. The SEM-EDX data are presented in Table S4. The O:F ratio in the reference KCrPO_4F was first estimated to be 3.97(8):1.03(8) and normalized to 4:1 yielding normalizing coefficients of 1.01 and 0.97 for O and F respectively. In the KTiPO_4F material the averaged O:F ratio was found to be 4.06(8):0.94(8), which after normalization gives 4.10(8):0.91(8) corresponding to the resulting formula of $\text{KTiPO}_{4.10(8)}\text{F}_{0.91(8)}$. Given the average value, the material seems to be slightly oxidized. However, it can be considered close to stoichiometric within the error window of determination.”

The discussion part has been updated with the following text:

“It was demonstrated that in many fluoride phosphates the fluorine-to-oxygen ratio in fluoride-phosphates plays a significant role in governing the electrochemical performance, which was observed and thoroughly studied for many vanadium-containing fluoride-phosphates. For those the situation is vastly complicated because both $\text{V}^{3+}/\text{V}^{4+}$ and $\text{V}^{4+}/\text{V}^{5+}$ (or $[\text{V}=\text{O}]^{2+}/[\text{V}=\text{O}]^{3+}$) redox transitions are typically electrochemically active⁵⁴. Deviation of the stoichiometric F:O ratio thus results in changing of electrochemical parameters (operating potential, de/intercalation mechanism, C-rates etc.)^{43,44,54-57}.

On the contrary, in case of titanium, shifting F:O ratio in KTiPO_4F implies appearance of Ti^{4+} which cannot be further oxidized upon depotassiation. Consequently, it means that in practice with higher O content the KTiPO_4F material becomes electrochemically inactive with Ti^{4+} content increasing. Approaching the KTiPO_4O composition, reversible electrochemical performance within the 2.0 – 4.2 vs. K^+/K potential region fully disappears. A recent paper demonstrated the applicability of KTiOPO_4 as an anode material for KIBs, showing the $\text{Ti}^{4+}/\text{Ti}^{3+}$ redox activity at an average potential of $\sim 0.8 \text{ V}$ ⁵⁸.

As it was shown by EELS and EPR measurements, the oxidation state of titanium in the material is 3+ and the BVS values for Ti are close to 3, which in accordance with the FTIR spectrum detecting no OH-groups capable to substitute fluorine and SEM-EDX analysis using a reference KCrPO_4F give us ground to assume a close to stoichiometric O:F ratio in KTiPO_4F . Overall, a more detailed investigation of the influence of the F:O ratio on the electrochemical performance might present a separate large and self-consistent study and thus can be published elsewhere.”

Overall, a more detailed investigation of the influence of the F:O ratio on the electrochemical performance might present a separate large and self-consistent study and thus can be published elsewhere.”

Question 3

The author considered the charge/discharge during 2.0-3.4V as solid-solution-like process, despite of clearly two oxidation peaks at 2.81 and 3.21 V for charge and two reduction peaks at 2.58 and 2.99 V for discharge, which demonstrate multiple phase transitions during the reaction. A close look in the operando XRD data did show the emergence of tiny peaks during the disappearance of (212) peak of the pristine sample. For example, some tiny ones at 29.7° can be observed when charged to 3.0V. A detailed discussion is needed to facilitate the understanding of structural evolution during the electrochemical reaction.

Response:

Please see the response to Question 2 of Reviewer 1, where detailed discussion on *operando* XRD is given.

Question 4

For the rate capability of the tested sample as shown in Figure 4c, the authors used different voltage windows, is there a special reason for such an operation? How about the Coulombic efficiency of the sample? The authors should compare at least the first 2 charge-discharge curves at 0.1 C of the samples, which are critical to understand the structural stability and transition of the electrode materials during cycling.

Response:

Since with increasing discharge currents the polarization of electrode potentials grows (which becomes really significant at high C-rates), to deliver more capacity on discharge the cathodic cut-off potentials at 5C and 10C measurements were slightly shifted to 1.8 V. The anodic potentials for all measurements were set to 4.2 V vs. K⁺/K. The coulombic efficiency of the sample at 2C and 5C rates are close to 99.5% as presented in Figure R6. The first 2 charge-discharge curves at 0.1 C compared in Figure. The discharge curve at second cycle coincides with that obtained at the first cycle confirming structural stability of our electrode.

Figure R6. Left: The first and second charge-discharge cycles at a C/20 rate. Inset: a dQ/dE differential plot for the second galvanostatic cycle at C/20. Right: Discharge capacities during extended cycling at 2C and 5C charge/discharge rate for 100 cycles and coulomb efficiency.

Figure R6 was added to the manuscript as a part of the modified Figure 2 with related comments throughout the text. Thank you.

References

- [1] E. Boivin, A. Iadecola, F. Fauth, J.-N. Chotard, C. Masquelier, and L. Croguennec, "Redox Paradox of Vanadium in Tavorite $\text{LiVPO}_4\text{F}_{1-y}\text{O}_y$," *Chem. Mater.*, vol. 31, no. 18, pp. 7367–7376, Sep. 2019.
- [2] J.-M. Ateba Mba, C. Masquelier, E. Suard, and L. Croguennec, "Synthesis and Crystallographic Study of Homeotypic LiVPO_4F and LiVPO_4O ," *Chem. Mater.*, vol. 24, no. 6, pp. 1223–1234, 2012.
- [3] K. Chihara *et al.*, "KVPO₄F and KVOPO₄ toward 4 volt-class potassium-ion batteries †," *Chem. Commun*, vol. 5208, p. 1, 2017.
- [4] M. Kim, S. Lee, and B. Kang, "Fast-Rate Capable Electrode Material with Higher Energy Density than LiFePO_4 : 4.2V LiVPO_4F Synthesized by Scalable Single-Step Solid-State Reaction," *Adv. Sci.*, vol. 3, no. 3, p. 1500366, 2016.
- [5] H. Kim *et al.*, "A New Strategy for High-Voltage Cathodes for K-Ion Batteries: Stoichiometric KVPO_4F ," *Adv. Energy Mater.*, vol. 8, no. 26, p. 1801591, Sep. 2018.
- [6] E. Boivin *et al.*, " $\text{LiVPO}_4\text{F}_{1-y}\text{O}_y$ Tavorite-Type Compositions: Influence of the Concentration of Vanadyl-Type Defects on the Structure and Electrochemical Performance," *Chem. Mater.*, vol. 30, no. 16, pp. 5682–5693, Aug. 2018.
- [7] R. Zhang *et al.*, "Safe, Low-Cost, Fast-Kinetics and Low-Strain Inorganic-Open-Framework Anode for Potassium-Ion Batteries," *Angew. Chemie Int. Ed.*, Sep. 2019.

Reviewers' comments:

Reviewer #1 (Remarks to the Author):

The manuscript is well revised according to reviewers' comments. After including the EELS and EPR results, the electrochemical mechanisms of the KTiPO_4F is clearly revealed and confirmed. Therefore, I recommend to accept this manuscript for publication in current stage.

Reviewer #2 (Remarks to the Author):

Well done. I believe that the present work will be one of the milestone to excavate a new compound for KIBs.

Reviewer #3 (Remarks to the Author):

I appreciate the efforts the authors have taken to address the questions I have raised. Here I still have a couple of major concerns before I can recommend an acceptance of this work in Nat Commun.

1. The fluorine-to-oxygen ratio turned out to play a critical role in determining the electrochemical properties in the prepared fluorophosphates. To make the data more convincing, it become necessary that the fluorine-to-oxygen ration be changed to show the composition-dependent performance.
2. The operando XRD data showed complex phase transitions during the electrochemical reaction. A detailed analysis on the multiple two-phase intercalations is needed so as to provide a clear understanding on the storage behavior of the KTiPO_4F .

Reviewer #1:

Comments to the author: The manuscript is well revised according to reviewers' comments. After including the EELS and EPR results, the electrochemical mechanisms of the KTiPO_4F is clearly revealed and confirmed. Therefore, I recommend to accept this manuscript for publication in current stage.

Response:

We are really grateful to the reviewer for his/her final opinion on the manuscript. Thank you.

Reviewer #2:

Comments to the author:

Well done. I believe that the present work will be one of the milestone to excavate a new compound for KIBs.

Response:

We really appreciate such a high assessment of our work. Thank you.

Reviewer #3:

Comments to the author:

I appreciate the efforts the authors have taken to address the questions I have raised. Here I still have a couple of major concerns before I can recommend an acceptance of this work in Nat Commun.

Question 1

The fluorine-to-oxygen ratio turned out to play a critical role in determining the electrochemical properties in the prepared fluorophosphates. To make the data more convincing, it become necessary that the fluorine-to-oxygen ration be changed to show the composition-dependent performance.

Response:

We agree that the fluorine-to-oxygen ratio (F:O) in fluoride-phosphates plays a significant role in governing the electrochemical performance. However, to correlate influence of the precise ratio of F:O upon electrochemical performance, in case of Ti^{3+} -based compounds is non-trivial. As the reviewer likely understands, the substitution of oxygen for fluorine in any Ti^{3+} -based materials leads

to the obvious deterioration of the electrochemical activity due to formation of Ti^{4+} ions, which are not prone to further oxidation within the reasonable range of applied electrode potentials. With increased oxygen content, the $\text{KTiPO}_{4+x}\text{F}_{1-x}$ material gradually becomes electrochemically inactive as we demonstrate in Figure R1, Left. Importantly, when the F:O ratio achieves 0.5:4.5 representing the $\text{KTiPO}_{4.5(1)}\text{F}_{0.5(1)}$ composition, the reversible specific capacity and the average potential significantly drop down, the 3.6 V plateau disappears (Figure R1, Right). Approaching the KTiPO_5 (KTiPO_4O , no fluorine) composition with Ti^{4+} being a dominant oxidation state, the material gets fully inactive within the 2.0 – 4.2 vs K^+/K potential range (Figure R1, left).

Figure R1. Left. CV curves for KTiPO_4F ($\text{KTiPO}_{4.1(1)}\text{F}_{0.9(1)}$), $\text{KTiPO}_{4.5(1)}\text{F}_{0.5(1)}$ and KTiPO_4O . Right. Galvanostatic curves for $\text{KTiPO}_{4.1(1)}\text{F}_{0.5(1)}$ and $\text{KTiPO}_{4.5(1)}\text{F}_{0.5(1)}$

We also emphasize that this experimental behavior of the $\text{KTiPO}_{4+x}\text{F}_{1-x}$ is in good agreement with the DFT+U calculations, where we considered two main intercalation steps, $\text{K}_1\text{TiPO}_{4+y}\text{F}_{1-y}/\text{K}_{0.5}\text{TiPO}_{4+y}\text{F}_{1-y}$ and $\text{K}_{0.5}\text{TiPO}_{4+y}\text{F}_{1-y}/\text{TiPO}_{4+y}\text{F}_{1-y}$ for $y = 0.125, 0.25,$ and 0.5 . Using one unit cell, the corresponding number of symmetrically non-equivalent F:O arrangements are 1, 3, and 4, respectively. For $y=0.25$ and 0.5 we chose the lowest energy F:O arrangement in $\text{K}_1\text{TiPO}_{4+y}\text{F}_{1-y}$, though, the type of F:O arrangement has little influence on the intercalation potential. To clarify, we show the potential dependence in Figure R2. As demonstrated for $\text{KTiPO}_4\text{F}/\text{K}_{0.5}\text{TiPO}_4\text{F}$ the influence of F:O ratio is insignificant (which is also observed on the galvanostatic curves, Figure R1, right), however, for the $\text{K}_{0.5}\text{TiPO}_4\text{F}/\text{K}_0\text{TiPO}_4\text{F}$ step, the potential starts upshifting rapidly as more F is replaced by O. For $y = 0.5$ all Ti in $\text{K}_{0.5}\text{TiPO}_{4.5}\text{F}_{0.5}$ is in the 4+ oxidation state and further extraction of

K results in the oxygen redox, increasing the potential up to 5.5 V (the 3.6 V plateau is not seen on the CV and charge/discharge curves for $\text{KTiPO}_{4.1(1)}\text{F}_{0.5(1)}$ (Figure R1).

Figure R2. DFT+U calculated average intercalation potentials for $\text{K}_1\text{TiPO}_{4+y}\text{F}_{1-y}/\text{K}_{0.5}\text{TiPO}_{4+y}\text{F}_{1-y}$ and $\text{K}_{0.5}\text{TiPO}_{4+y}\text{F}_{1-y}/\text{TiPO}_{4+y}\text{F}_{1-y}$ transitions at different F:O ratios ($y = 0, 0.125, 0.25,$ and 0.5).

Therefore, the fluorine content should be kept closest to the KTiPO_4F stoichiometry for realizing high-voltage performance of the material and it is experimentally difficult to synthesize and characterize such small substitutions of F for O to reflect the change in materials behavior. **We have updated the manuscript and supporting information with the data given above to detail the sensitivity to F:O ratio.**

Question 2

The operando XRD data showed complex phase transitions during the electrochemical reaction. A detailed analysis on the multiple two-phase intercalations is needed so as to provide a clear understanding on the storage behavior of the KTiPO_4F .

Response:

To address this comment, we have provided additional synchrotron X-ray powder diffraction (SXPDP) data to shed more light on the multiple two-phase intercalations, which are interesting to many readers from the storage behavior mechanism standpoint.

We note that a general view of the diffraction patterns evolution (Figure R3, A) clearly shows multiple regions with distinct two-phase or solid-solution transitions and indicates that these processes are symmetric and fully reversible within one charge-discharge cycle as also demonstrated by *ex situ* XRD patterns (Figure R3, C). At a closer look at the intensity map of selected two-theta sections (Figure R4, A) one can observe three two-phase transitions accompanied by appearance/disappearance of specific reflections which is connected with changes in the unit cell symmetry, and one solid-solution region with no distinct changes in symmetry. In addition it is worth noting that the orthorhombic symmetry (SG # 33 $Pna2_1$) of the initial KTiPO_4F material was validated by electron diffraction.

K deintercalation in KTiPO_4F starts with a two-phase transition as indicated by disappearance of the 212 (Figure R4, A, top) and 323 reflections possibly introducing C ($hkl: h+k = 2n$) or A ($hkl: k+l = 2n$) centering to the unit cell. This centering can result from switching to a higher or lower symmetry. Minimal supergroups for $Pna2_1$ with the corresponding centering conditions are $Ccm2_1$ (non-standard setting) or $Ama2$ (origin at $0 \frac{1}{4} 0$). However, these space groups fail to describe the existing peaks on the diffraction patterns due to extra extinction conditions characteristic to these supergroups thus requiring a monoclinic distortion leading to Cc ($Cm, C2/c$) or Ac ($Am, A2$) space groups. Lowering symmetry for the initial $Pna2_1$ space group from orthorhombic to monoclinic can directly result in the Cc subgroup. At this point due to insufficient resolution of the 2D detector resulting in large peak halfwidths on SXPD patterns, making an explicit choice of the space group becomes complicated. Nevertheless, among the mentioned space groups the best fit of diffraction pattern profile is achieved with the Cm space group which is taken for cell parameters refinement. Moreover, various K ions ordering during extraction may result even into lower symmetries with larger unit cells as evidenced by DFT for intermediate compositions ($\text{K}_{0.875}, \text{K}_{0.75}$ etc.) converged to monoclinic and triclinic structures with larger unit cells. Further K deintercalation tentatively proceeds via a solid-solution mechanism as no changes in symmetry can be found. This region is characterized by a gradual shift of peaks positions and a slope behavior of the corresponding voltage curve, and presumably ends at 3.1 V (on charge) followed by another two-phase transition. The symmetry choice for the third and fourth two-phase transition was guessed from the DFT relaxation of the corresponding structures. A short plateau centered at 3.2 V specifies the symmetry change back to $Pna2_1$ with appearance of the 210 and 410 reflections at 9.3° and 14.3° 2θ respectively belonging to this space group which further disappear designating a transition to the $Pnan$ space group where these reflections are forbidden (in $Pnan$ hko :

$h+k = 2n$). The crystal symmetry of the charged material well satisfies the extinction conditions of the centrosymmetric $Pnan$ space group (the hko reflections with $h+k \neq 2n$ are absent due to the n -glide perpendicular to the c axis). Furthermore, the crystal structures of charged electrodes of the isostructural materials ($KVPO_4F$ and $KFeSO_4F$) also adopt this symmetry which was earlier demonstrated experimentally^{41,50}.

The phase cell volume evolution associated with the above-mentioned transitions and corresponding charge curve are given in Figure R4, B. It should be noted that the two-phase transitions are also clearly seen on the dQ/dE curve (Figure R4, C) as reversible peaks vs. the solid-solution region with no discernable peaks.

Figure R3. A) Diffraction patterns of the *in operando* SXP of $KTiPO_4F$ in K cell in the $6\text{--}17^\circ$ 2θ range ($\lambda = 0.7239 \text{ \AA}$) and corresponding charge-discharge profile (B). Indexed reflections are denoted. The highlighted dark-red XRD patterns show tentative boundaries of the two-phase (2P) or solid solution (SS) deintercalation regimes on charge. The discharge behavior of $KTiPO_4F$ can be considered symmetric. The asterisk sign (*) designates steady reflections belonging to cell components. C) Selected regions of the *ex situ* XPD patterns of electrodes recovered at various potentials (charge/discharge at C/20 rate, hold for 20 hrs. at every potential).

Figure R4. A) Magnified regions of the *in operando* SXPd intensity map: phase transformations. White and black dotted lines designate reflections related to particular phases. 2P – two-phase region, SS – solid-solution region. Vertical dashed lines show tentative borders of 2P and SS regions. B) Cell volume evolution on charge refined from *in operando* SXPd data for $\text{K}_x\text{TiPO}_4\text{F}$ with related charge galvanostatic curve. C) dQ/dE plot showing potential ranges for solid-solution and multiple two-phase deintercalation regions.

To make this all the more clear we have updated the manuscript and SI to detail this complicated mechanistic behavior.

- [1] S. S. Fedotov *et al.*, “ AVPO_4F (A = Li, K): A 4 v Cathode Material for High-Power Rechargeable Batteries,” *Chem. Mater.*, vol. 28, no. 2, pp. 411–415, 2016.
- [2] T. Hosaka, T. Shimamura, K. Kubota, and S. Komaba, “Polyanionic Compounds for Potassium-Ion Batteries,” *Chem. Rec.*, vol. 19, no. 4, pp. 735–745, Apr. 2019.

REVIEWERS' COMMENTS:

Reviewer #3 (Remarks to the Author):

I am ok with the arguments the authors provided in the rebuttal letter. I 'd like to recommend an acceptance of this manuscrip for its publication in Nature Communications.

Reviewer #3:

Comments to the author: I am ok with the arguments the authors provided in the rebuttal letter. I'd like to recommend an acceptance of this manuscript for its publication in Nature Communications.

Response:

We are grateful to the reviewer for the positive final opinion on our manuscript.